TRANSPARENT PROCESS OPEN ACCESS Life Science Alliance



# Elevated glycolysis imparts functional ability to CD8⁺ T cells in HIV infection

AKM Nur-ur Rahman[1], Jun Liu[1], Shariq Mujib[2], Segen Kidane[2], Arman Ali[1], Steven Szep[1], Carrie Han[1], Phil Bonner[1], Michael Parsons[3], Erika Benko[4], Colin Kovacs[4], Feng Yun Yue[1], Mario Ostrowski[1,2,5,6]

**The mechanisms inducing exhaustion of HIV-specific CD8⁺ T cells are not fully understood. Metabolic programming directly influences T-cell differentiation, effector function, and memory. We evaluated metabolic profiles of ex vivo CD8⁺ T cells in HIV-infected individuals. The baseline oxygen consumption rate of CD8⁺ T cells was elevated in all infected individuals and CD8⁺ T cells were working at maximal respiratory capacity. The baseline glycolysis rate was enhanced only during early untreated HIV and in viral controllers, but glycolytic capacity was conserved at all stages of infection. CD8⁺ T-cell mTOR activity was found to be reduced. Enhanced glycolysis was crucial for HIV-specific killing of CD8⁺ T cells. CD8⁺ T-cell cytoplasmic GAPDH content was reduced in HIV, but less in early infection and viral controllers. Thus, CD8⁺ T-cell exhaustion in HIV is characterized by reduced glycolytic activity, enhanced OXPHOS demands, dysregulated mTOR, and reduced cytoplasmic GAPDH. These data provide potential metabolic strategies to reverse CD8⁺ T-cell dysfunction in HIV.**


## Introduction

Cytotoxic CD8⁺ T cells are immune sentinels responsible for eliminating virus-infected and cancer cells. Resting CD8⁺ T cells have low energy requirements, which are met primarily by oxidative phosphorylation (OXPHOS) to generate ATP, and by catabolic metabolism to break down and recycle internal molecules that supply biochemical substrates to sustain homeostatic survival (Pearce et al, 2013; Pearce & Pearce, 2013). Once activated, CD8⁺ T cells require extraordinary amounts of energy and biochemical substrates, which rival that of cancer cells, to facilitate expansive cellular division and transform into effector cells to exert functional control of target cells (Pearce et al, 2009; Jones et al, 2017). Activated CD8⁺ T cells up-regulate aerobic glycolysis to generate ATP and, in addition, switch to anabolic metabolism by actively transporting nutrients into the cell to provide biochemical substrates to sustain a proliferative burst and gain effector function differentiation (Gubser et al, 2013; Gerriets et al, 2015; Cunningham et al, 2018). Once the target cells are eliminated, antigen (Ag)-specific CD8⁺ T cells undergo contraction and a smaller pool of these Ag-specific CD8⁺ T cells live on as memory cells primed for immediate immune response to re-challenge (Hosking et al, 2014). Recent work has highlighted that metabolic programming itself influences CD8⁺ T-cell proliferation, differentiation, effector function, persistence, and survival into memory (van der Windt et al, 2013; Chang et al, 2013; Okoye et al, 2015; Bengsch et al, 2016; Schurich et al, 2016). Yet, the widespread presence of chronic viral infection and cancer indicate that CD8⁺ T-cell–mediated immune response fails quite often.

Globally 38 million individuals are currently living with HIV (World Health Organization, 2020). During early HIV infection, which occurs between the first 6 and 9 mo of infection, the HIV-specific CD8⁺ T cells undergo massive proliferation in response to high HIV viremia and are functionally capable of partial viral control towards the individual viral setpoint, whereby viremia persists into chronic phase without combination antiretroviral therapy (cART) (Ogg et al, 1998; Shankar et al, 2000; Soghoian et al, 2012; Du et al, 2016). The failure of the host immune system to contain HIV infection during the chronic phase is related to the well documented, step-wise progressive functional impairment of HIV specific CD8⁺ T cells, known as T-cell exhaustion (Day et al, 2006; Buggert et al, 2014). CD8⁺ T-cell exhaustion persists despite effective cART, making immunotherapies problematic. These exhausted CD8⁺ T cells express the classical exhaustions markers PD-1, CTLA-4, LAG-3, TIM-3, and TIGIT among others; and skewed maturation phenotypes (Blackburn et al, 2009; Yamamoto et al, 2011; Passaes et al, 2020). Surprisingly, a minority of HIV-infected individuals, known as viral controllers (VCs), are able to control the virus without cART intervention with robust responses against HIV-infected cells from HIV specific CD8⁺ T cells and maintain healthy CD4⁺ T-cell counts (Deeks & Walker, 2007; Thèze et al, 2011). The CD8⁺ T cells of VCs can maintain their functionality over a longer duration without being exhausted (Betts et al, 2006; Migueles et al, 2008; Hersperger et al, 2010). In addition,

[1]Deparment of Medicine, University of Toronto, Toronto, Canada [2]Institute of Medical Sciences, University of Toronto, Toronto, Canada [3]Flow Cytometry Facility, Lunenfeld Tanenbaum Research Institute, Mount Sinai Hospital, Toronto, Canada [4]Maple Leaf Medical Clinic, Toronto, Canada [5]Deparment of Immunology, University of Toronto, Toronto, Canada [6]Keenan Research Centre for Biomedical Sciences of St. Michael's Hospital Toronto, Toronto, Canada

Correspondence: mario.ostrowski@gmail.com

 

Angin et al (2019), using a transcriptiomics approach, compared immunometabolic profiles of CD8⁺ T cells in VCs and cART-treated individuals and observed greater metabolic plasticity of CD8⁺ T cells in VCs (Angin et al, 2019).

Here, we further investigated metabolic pathophysiological processes in human ex vivo CD8⁺ T cells during various stages of HIV infection. HIV infection presents us with the opportunity of investigating the metabolic reprogramming that CD8⁺ T cells undergo during the different phases of HIV disease progression. During early HIV infection, HIV specific CD8⁺ T cells are functionally active and able to partially control viremia, but soon become functionally impaired during the chronic phase. We expect the metabolic profiles of CD8⁺ T cells during chronic infection to be most closely associated with the exhausted phenotype. Our goal is to develop a metabolic model of T-cell functionality and exhaustion in HIV infection and to investigate whether metabolic interventions can enhance and/or rescue functionality of HIV-specific CD8⁺ T cells.

## Results

### Metabolic status of ex vivo CD8⁺ T cells after overnight rest

To establish how HIV infection alters global cellular metabolism, we investigated the metabolic status of ex vivo CD8⁺ T cells from HIV-infected individuals from three distinct clinical stages, cART-naïve and treated: (1) early (acquired within 6 mo); (2) chronic progressive (infected > 1 yr, CD4⁺ T-cell count of <500/μl); and (3) VCs (infected > 1 yr, plasma viral load <5,000 copies/ml, CD4⁺ T-cell count of >500/μl, not on cART). Early HIV-infected individuals were sampled at presentation and then after 1 yr on cART (HIV plasma viral load undetectable in all). We negatively isolated total CD8⁺ T cells from HIV-infected and HIV-uninfected control donors, rested them overnight, and then measured their bioenergetic profiles in extracellular flux at basal state, and after the addition of oligomycin (to block ATP synthesis), FCCP (to uncouple ATP synthesis from electron transport chain, ETC), and rotenone and antimycin A (to inhibit complex I and III of ETC) (Fig S1A and B). Oxygen consumption rate (OCR) reflects OXPHOS, whereas extracellular acidification rate (ECAR), caused by lactic acid secretion in the extracellular space, reflects the glycolysis rate.

#### *HIV-uninfected versus HIV-early infection*

The baseline OCR, indicating the basal $O_2$ consumption rate of homeostatic maintenance of CD8⁺ T cells, and ATP linked OCR, indicating the rate of $O_2$ consumption to generate ATP, were similar for CD8⁺ T cells from HIV-uninfected and early HIV-infected cART–naïve donors (see Fig 1A–C and Table S1). The spare capacity OCR, previously reported to be directly linked to memory generation (van der Windt et al, 2012), was higher by almost twofold in CD8⁺ T cells from early HIV-infected cART–naïve than in HIV-uninfected controls (Fig 1A and D and Table S1). In contrast, the baseline, ATP-linked, and spare capacity OCRs were significantly lower in CD8⁺ T cells from early HIV-infected cART–treated donors than in HIV-uninfected control donors (Fig 1B–D and Table S1). Of note, the early-treated groups were the same individuals studied before treatment.

The baseline ECAR, indicating the basal glycolysis rate to generate ATP for homeostatic maintenance of CD8⁺ T cells, maximal ECAR, indicating the maximal rate at which glycolysis can be performed when the mitochondria are shut down, and glycolytic capacity ECAR, indicating the energy reserve that CD8⁺ T cells can use to generate ATP in times of mitochondrial stress, were significantly higher in CD8⁺ T cells from early HIV-infected cART–naïve donors than that from CD8⁺ T cells from HIV-uninfected donors (Fig 1E–H and Table S1). For the same early HIV-infected individuals who were on cART for at least 1 yr, the baseline and maximal ECARs reduced to that of HIV-uninfected donors but glycolytic capacity ECAR was significantly depressed (Fig 1E–H and Table S1).

#### *HIV-uninfected versus HIV chronic infection*

In contrast to untreated early HIV, CD8⁺ T-cell OCR from both untreated and cART-treated chronic HIV, were mildly elevated from uninfected donors. OCR capacity was twofold higher than that of CD8⁺ T cells from HIV-uninfected control donors, similar to untreated early (Fig 2A–D and Table S1). In contrast to early HIV-infected cART–naïve donors, the baseline ECAR of CD8⁺ T cells of chronic HIV-infected cART–naïve donors was not elevated, but maximal ECAR trended higher. The glycolytic capacity ECAR of CD8⁺ T cells of chronic HIV-infected cART–naïve was not higher than uninfected, and chronic cART treatment showed a trend to decrease in glycolytic capacity ECAR (Fig 2G and H and Table S1).

#### *HIV-uninfected versus viral controller*

The baseline and ATP-linked OCR were significantly higher in CD8⁺ T cells from HIV VCs than from uninfected donors (Fig 3A–C and Table S1). The spare capacity OCR trended to be higher in VCs (Table S1 and Fig 3D). The baseline ECAR was significantly increased in CD8⁺ T cells from HIV VCs (Fig 3E and F and Table S1), the maximal ECAR was non-significantly increased and glycolytic capacity similar compared to that of CD8⁺ T cells from HIV-uninfected donors (Fig 3G and H and Table S1).

### Metabolic status of ex vivo CD8⁺ T cells after overnight activation

To further investigate how HIV infection alters the metabolic response of global CD8⁺ T cells when recalled to respond, we activated ex vivo CD8⁺ T cells from donors overnight with plate-bound anti-human CD3 (1 μg/ml) and anti-human CD28 (5 μg/ml) antibodies. We deemed changes in metabolic profiles based on OCR (OXPHOS) and ECAR (glycolysis) readings would reflect in vivo metabolic transformation that CD8⁺ T cells undergo before entering the proliferation cycle. We compared CD3/CD28–activated CD8⁺ T-cell profiles with their corresponding resting CD8⁺ T cells and with activated CD8⁺ T cells from HIV-uninfected individuals (Table S1 and summary in Fig 4).

#### *OCR (OXPHOS) assessment*

As expected, the baseline, ATP-linked, and spare capacity OCRs of CD8⁺ T cells from HIV-uninfected donors increased significantly after overnight CD3/CD28 activation when compared to overnight resting cells (Table S1 and Fig S2A–E). CD8⁺ T cells from acute-treated individuals also behaved similarly. Surprisingly, for CD8⁺ T cells from cART naive early, naive, and cART-treated chronics,

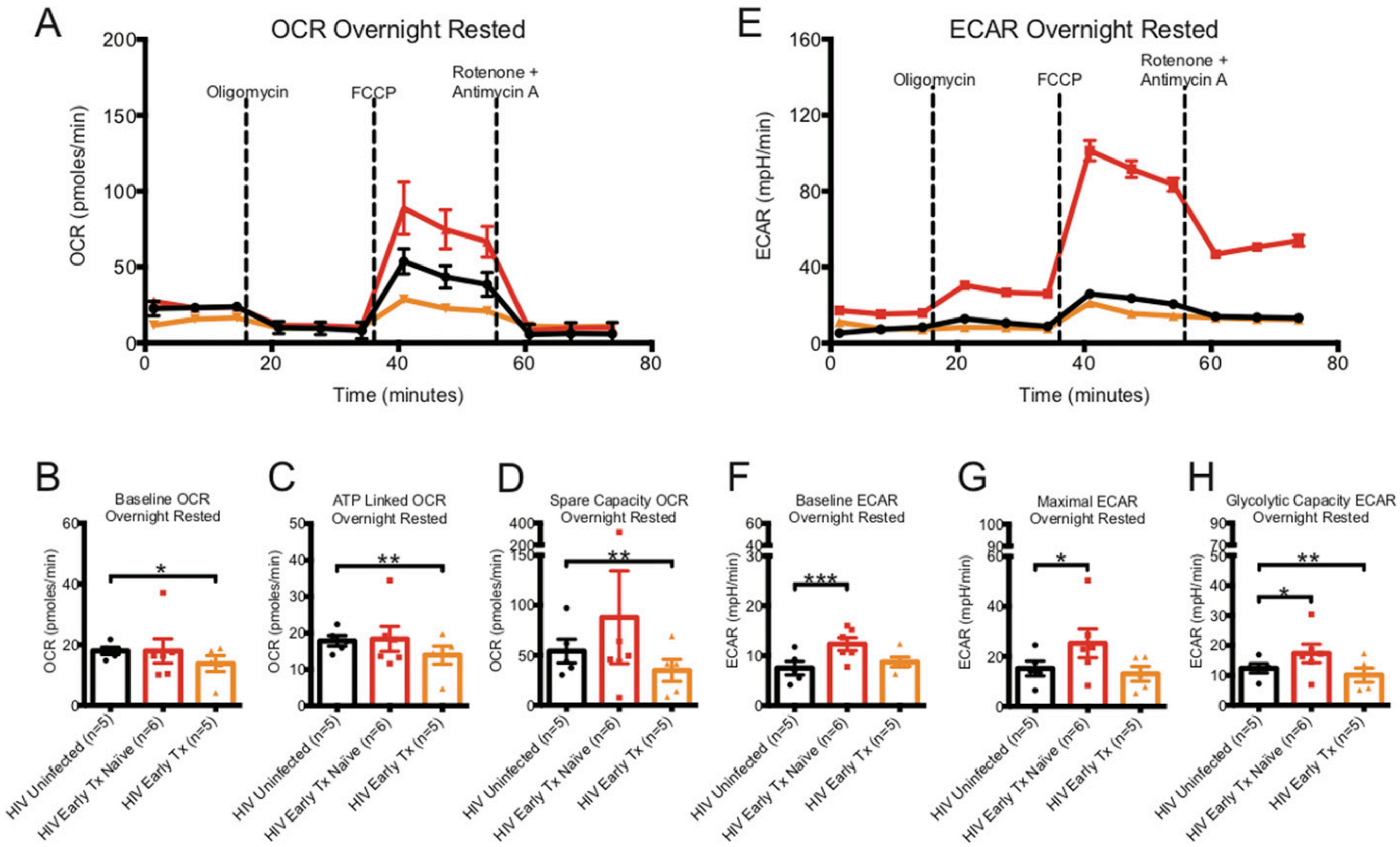


**Figure 1. Oxygen consumption rate (OCR) and ECAR metabolic profiles of ex vivo overnight rested or activated CD8⁺ T cells from HIV early–infected cART–naïve and cART–treated donors.**

**(A)** illustrates a representative OCR profile of overnight-rested $CD8^+$ T cells taken ex vivo from HIV-uninfected (black line), HIV-early cART–naïve (red line), and HIV early–cART–treated (orange line) donors. These are prospectively matched samples from the same donors before and after being on cART treatment for at least 1 yr. **(B, C, D)** Hollow bars represent baseline, ATP-linked, and spare capacity OCRs of overnight-rested $CD8^+$ T cells from HIV-uninfected (black line, n = 5), HIV-early cART–naïve (red line, n = 6), and HIV-early cART–treated (orange line, n = 5) donors. **(E)** illustrates a representative ECAR profile of ex vivo overnight-rested $CD8^+$ T cells from HIV-uninfected (black line), HIV-early cART–naïve (red line), and HIV-early cART–treated (orange line) donors. **(F, G, H)** Hollow bars represent baseline, maximal, and glycolytic capacity ECARs of overnight-rested $CD8^+$ T cells from HIV-uninfected (black line, n = 5), HIV-early cART–naïve (red line, n = 6), and HIV-early cART–treated (orange line, n = 5) donors. Data are representative of at least three technical repeats. * indicates $P < 0.05$, ** $P < 0.01$, and *** $P < 0.001$ by unpaired two tailed nonparametric Mann–Whitney test. Error bars are mean ± SEM.

although the baseline OCR increased, the spare OCR capacity was unchanged in comparison to their resting counterparts. $CD8^+$ T cells from VCs were unique in that activation resulted in no change of baseline and spare OCR compared to their resting cell counterparts (Table S1 and Fig S2A–E).

When comparing CD3/CD28 stimulated $CD8^+$ T cells from uninfected individuals to the HIV groups, we saw greater spare OCR capacity only in $CD8^+$ T cells from naïve and cART-treated chronic infection but lower spare OCR in treated early and VCs (Table S1 and Fig S2A–E).

### *ECAR (glycolysis) assessment*

$CD8^+$ T cells from HIV-uninfected and all HIV-infected groups had significantly higher levels of baseline, maximal, and glycolytic capacity ECARs after overnight CD3/CD28 activation when compared with their resting counterparts (Table S1 and Fig S3A–E).

When comparing activated $CD8^+$ T cells from uninfected individuals, baseline, maximal, and glycolytic capacity ECARs were greater in early cART-naïve HIV but were decreased in the early cART-treated group and were at the same level for chronics and VCs (Table S1 and Fig S3A–E).

### Summary of ex vivo $CD8^+$ T cells metabolic data (see Fig 4)

To summarize, ex vivo overnight rested $CD8^+$ T cells from HIV-infected individuals generally had elevated baseline mitochondrial respiration (OXPHOS) and enhanced mitochondrial reserve capacity compared with HIV-uninfected donors (Fig 4A). The only exception to this was that of cART-treated early individuals, who had reduced baseline OXPHOS and mitochondrial reserve capacity compared with HIV-uninfected donors (Fig 4A and B). $CD8^+$ T cells from early-HIV cART–naïve, naïve, and cART-treated chronic HIV, and VCs were functioning at maximal respiratory capacity because we did not observe enhancement in OXPHOS spare capacity after CD3/CD28 stimulation (Fig 4C). The latter was not observed in early cART-treated HIV or $CD8^+$ T cells from uninfected individuals. Ex vivo rested $CD8^+$ T cells from early HIV-infected cART–naïve and VCs were the only two groups that showed enhanced baseline glycolytic activity compared with uninfected individuals (Fig 4A). Thus, both

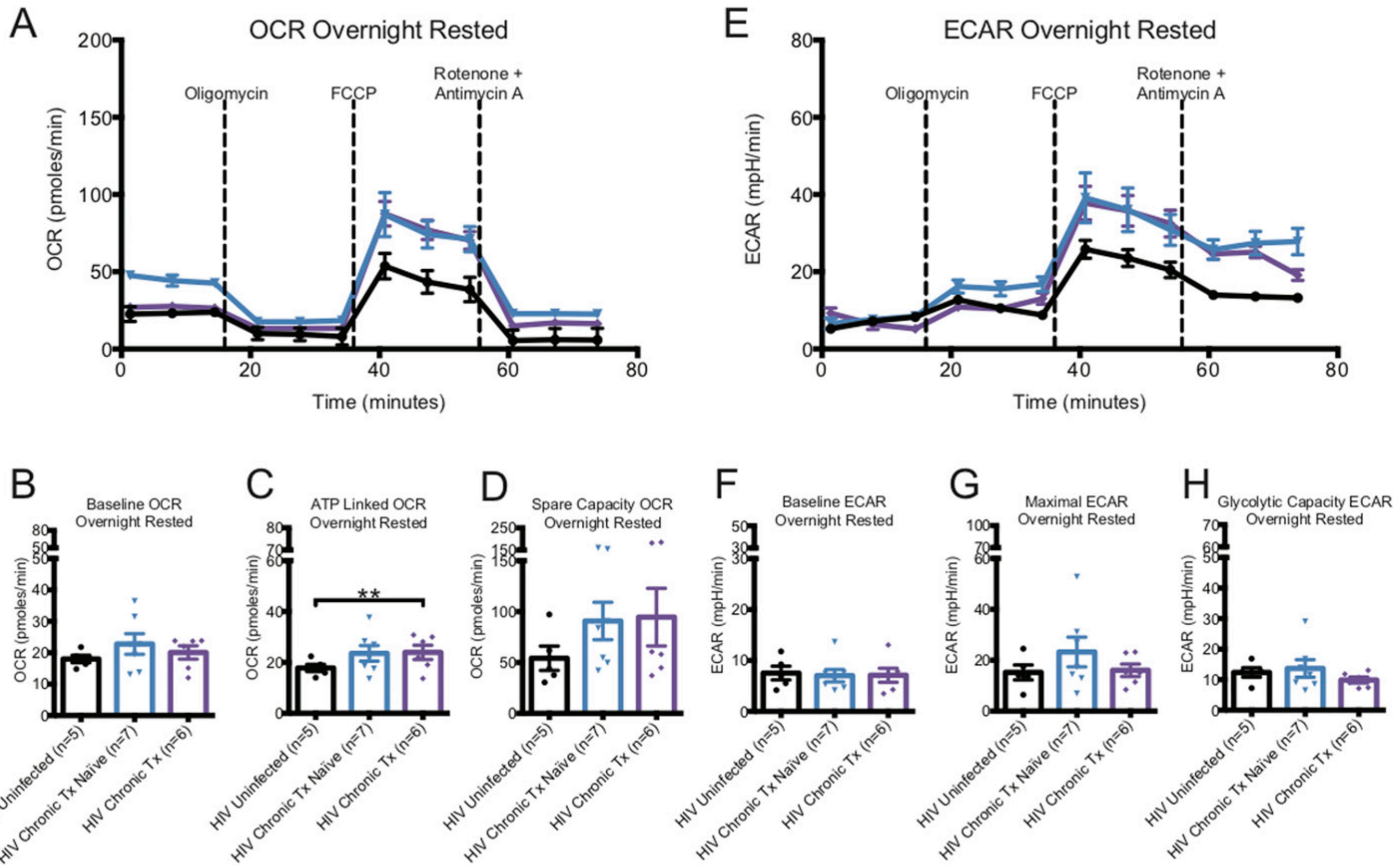


**Figure 2. Oxygen consumption rate (OCR) and ECAR metabolic profiles of ex vivo overnight-rested or activated CD8⁺ T cells from HIV-chronic cART–naïve and chronic CART–treated donors.**

**(A)** illustrates a representative OCR profile of overnight-rested CD8⁺ T cells from HIV-uninfected (black line), HIV-chronic cART–naïve (blue line), and HIV-chronic cART–treated (purple line) donors. **(B, C, D)** Hollow bars represent baseline, ATP-linked, and spare capacity OCRs of overnight-rested CD8⁺ T cells from HIV-uninfected (black line, n = 5), HIV-chronic cART–naïve (blue line, n = 7), and HIV-chronic cART–treated (purple line, n = 6) donors. **(E)** illustrates a representative ECAR profile of overnight-rested CD8⁺ T cells from HIV-uninfected (black line), HIV-chronic cART–naïve (blue line), and HIV-chronic cART–treated (purple line) donors. **(F, G, H)** Hollow bars represent baseline, maximal, and glycolytic capacity ECARs of overnight-rested CD8⁺ T cells from HIV-uninfected (black line, n = 5), HIV-chronic CART–naïve (blue line, n = 7), and HIV-chronic cART–treated (purple line, n = 6) donors. Data are representative of at least three technical repeats. * indicates $P < 0.05$, ** $P < 0.01$, and *** $P < 0.001$ by unpaired two-tailed nonparametric Mann–Whitney test. Error bars are mean ± SEM.

early HIV-infected cART–naïve and VCs had both elevated baseline glycolysis and spare capacity OCR (Fig 4A) when compared with uninfected individuals. Despite persistent viremia, CD8⁺ T cells from chronic HIV-infected cART–naïve phase more resembled glycolytic metabolic profiles of CD8⁺ T cells from HIV-uninfected donors (Fig 4A and B). These features suggest that CD8⁺ T cells during chronic HIV infection failed to maintain the elevated baseline glycolysis rates observed during the early HIV-infected cART–naïve phase but were maximizing the use of spare capacity OCRs (Fig 4A).

### mTORC signaling and metabolic marker expression

Mechanistic target of rapamycin complex (mTORC) is the signaling hub that integrates information from environmental cues, nutrient availability, oxygen level, and energy levels with signal 1 and signal 2 of T-cell activation to properly activate CD8⁺ T cells (Yang & Chi, 2012; Donnelly et al, 2014; Hukelmann et al, 2016; Tkachev et al, 2017; Bantug et al, 2018). Because activation of CD8⁺ T cells is sustained by meeting the increased metabolic demands, we looked to see if CD8⁺ T cells had altered mTORC1 activation levels, as determined by pS6S240/244 expression. Surprisingly, the activity level of mTORC1 in ex vivo CD8⁺ T cells from early HIV-infected cART–naïve and chronic HIV-infected cART-treated donors were half of that of CD8⁺ T cells from HIV-uninfected donors. However, after being activated with plate-bound anti-human CD3/CD28 antibodies for 5 h, the mTORC1 activity was similar in the three groups (Fig 5A). Thus, ex vivo CD8⁺ T cells from HIV-infected individuals had reduced mTORC1 signaling activity compared with those from HIV-uninfected individuals which can be reversed with maximal stimulation through the TCR.

Expression of the metabolic markers CD98 and CD71 on the surface of CD8⁺ T cells was examined. CD98 is a type II single-pass transmembrane heavy chain heterodimer (SLC3A2 and SLC7A5) that forms a large neutral amino acid transporter on the surface of the cell necessary for nutrient transport. CD98 binds to cytoplasmic tails of integrin-$\beta$ chains and mediates adhesive signals that control cell spreading, survival, and growth (reviewed in Cantor and Ginsberg [2012]). CD98 has been shown to be induced by mTORC1 signaling (Hukelmann et al, 2016; Salzberger et al, 2018; Wang et al, 2018). Compared to CD8⁺ T cells from HIV-uninfected donors, CD98 expression was significantly decreased in CD8⁺ T cells from early

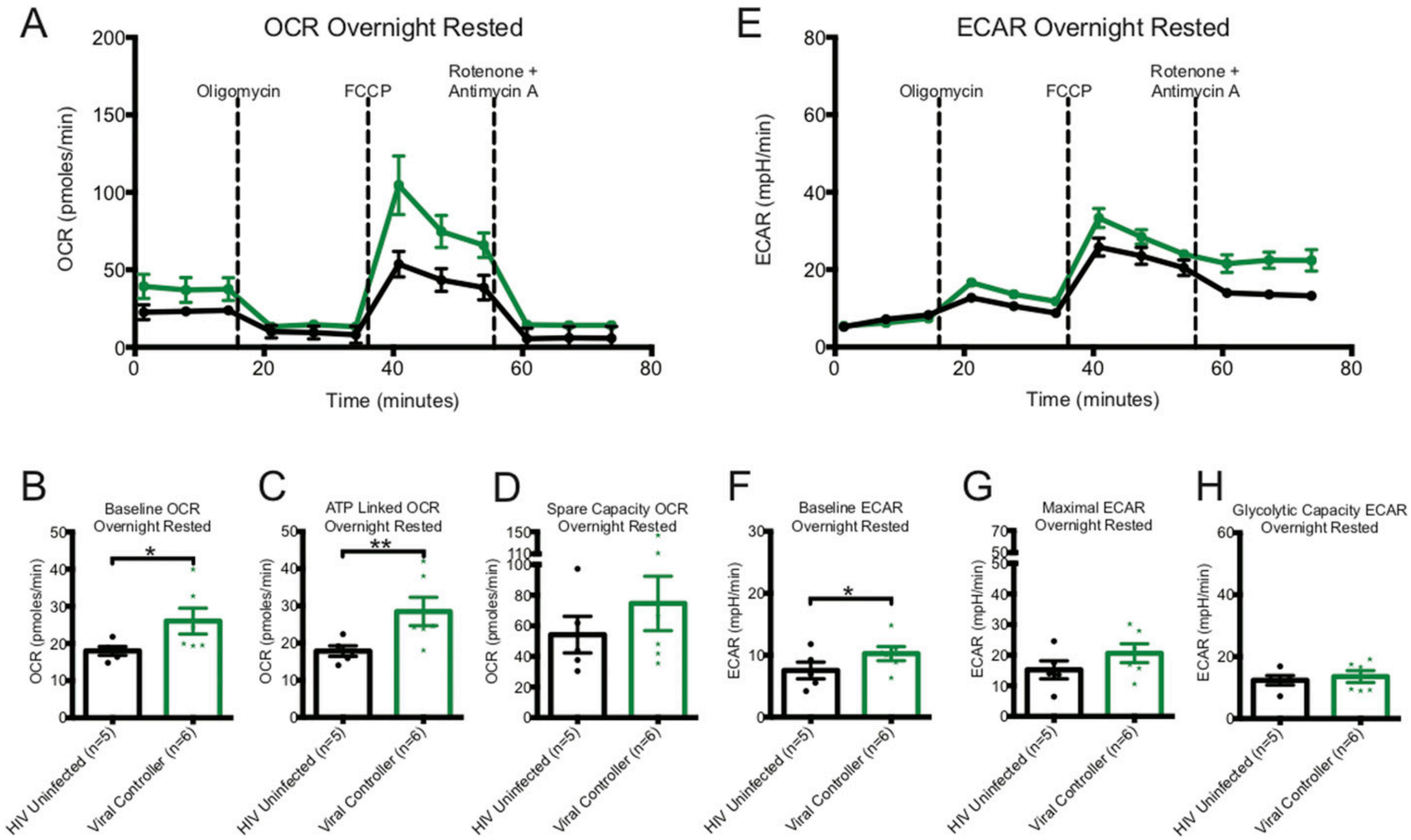


**Figure 3. Oxygen consumption rate (OCR) and ECAR metabolic profiles of overnight-rested or activated CD8⁺ T cells from viral controllers (VCs).**
**(A)** illustrates a representative OCR profile of overnight rested CD8⁺ T cells from HIV-uninfected (black line) and VC (green line) donors. **(B, C, D)** Hollow bars represent baseline, ATP-linked, and spare capacity OCRs of overnight-rested CD8⁺ T cells from HIV-uninfected (black line, n = 5), and VC (green line, n = 6) donors. **(E)** illustrates a representative ECAR profile of overnight rested CD8⁺ T cells from HIV-uninfected (black line) and VC (green line) donors. **(F, G, H)** Hollow bars represent baseline, maximal, and glycolytic capacity ECARs of overnight-rested CD8⁺ T cells from HIV-uninfected (black line, n = 5) and VC (green line, n = 6) donors. Data are representative of at least three technical repeats. * indicates $P < 0.05$, ** $P < 0.01$, and *** $P < 0.001$ by unpaired two-tailed nonparametric Mann–Whitney test. Error bars are mean ± SEM.

HIV-infected cART–naïve individuals (91.8% ± 1.8% versus 76.0% ± 2.6%, ** $P = 0.0015$) which was also observed in chronic HIV-infected cART–treated phase (91.8% ± 1.8% versus 79.8% ± 2.5%, ** $P = 0.003$) (Fig 5B). The amount of CD98 (MFI) expressed on CD8⁺ T cells was decreased in early HIV-infected cART–naïve individuals (565.2 ± 17.7 versus 599.7 ± 30.2) and this was more marked in chronic HIV-infected cART–treated infection (533.2 ± 9.9 versus 599.7 ± 30.2, * $P = 0.0447$) (Fig 5C).

CD71 is a type II transmembrane homodimeric glycoprotein responsible for cellular iron uptake via internalization of iron-loaded transferrin. Activated T cells express CD71 on their surface to meet the cellular demand for iron to sustain proliferation. CD71 is an early activation marker linked to proliferation (Motamedi et al, 2016), but is only partially regulated by mTORC1 activation because rapamycin does not inhibit its expression (Hukelmann et al, 2016; Salzberger et al, 2018; Wang et al, 2018). CD71 was expressed minimally on CD8⁺ T cells of HIV-uninfected donors (1.3% ± 0.3%) (Fig 5D) but was increased 10-fold during early HIV-infected cART–naïve phase (10.4% ± 1.2%, **** $P < 0.0001$). CD71 was reduced during chronic HIV-infected cART–treated phase (2.1% ± 0.6%) but remained at almost twofold above that of HIV-uninfected donors (Fig 5D). The amount (MFI) of CD71 expressed on the surface of CD8⁺ T cells from early HIV-infected cART–naïve donors increased significantly and was also maintained during chronic HIV-infected cART–treated phase despite reduced frequencies (Fig 5E). Thus, although CD8⁺ T cells were activated during early and chronic HIV infection, even during cART, this activation state is not associated with enhanced mTORC1 signaling activity ex vivo but suggests a counter-regulatory effect during HIV infection, or mTORC1 dysregulation.

## mTORC1 manipulation fails to improve ex vivo CD8⁺ T-cell function

Recently, it has been shown in the lymphocytic choriomeningitis virus infection model that early exhausted CD8⁺ T cells had elevated mTOR activity (Bengsch et al, 2016). Interestingly, it has been shown that CD4⁺ regulatory T cells ($T_{Regs}$) maintained their hyporesponsiveness by maintaining higher mTORC1 activity, which surprisingly was reversed with transient blockade of mTORC1 by rapamycin, suggesting an oscillatory switch in mTORC1 that control CD4⁺ $T_{Regs}$ function (Porcellini et al, 2010). Although we did not see enhanced mTOR signaling in ex vivo CD8⁺ T cells in HIV, we tested if mild inhibition of mTORC1 signaling in CD8⁺ T cells from chronic HIV-infected donors would be able to restore their functional capacity (Fig 5A). It has been shown that mild mTORC1 inhibition

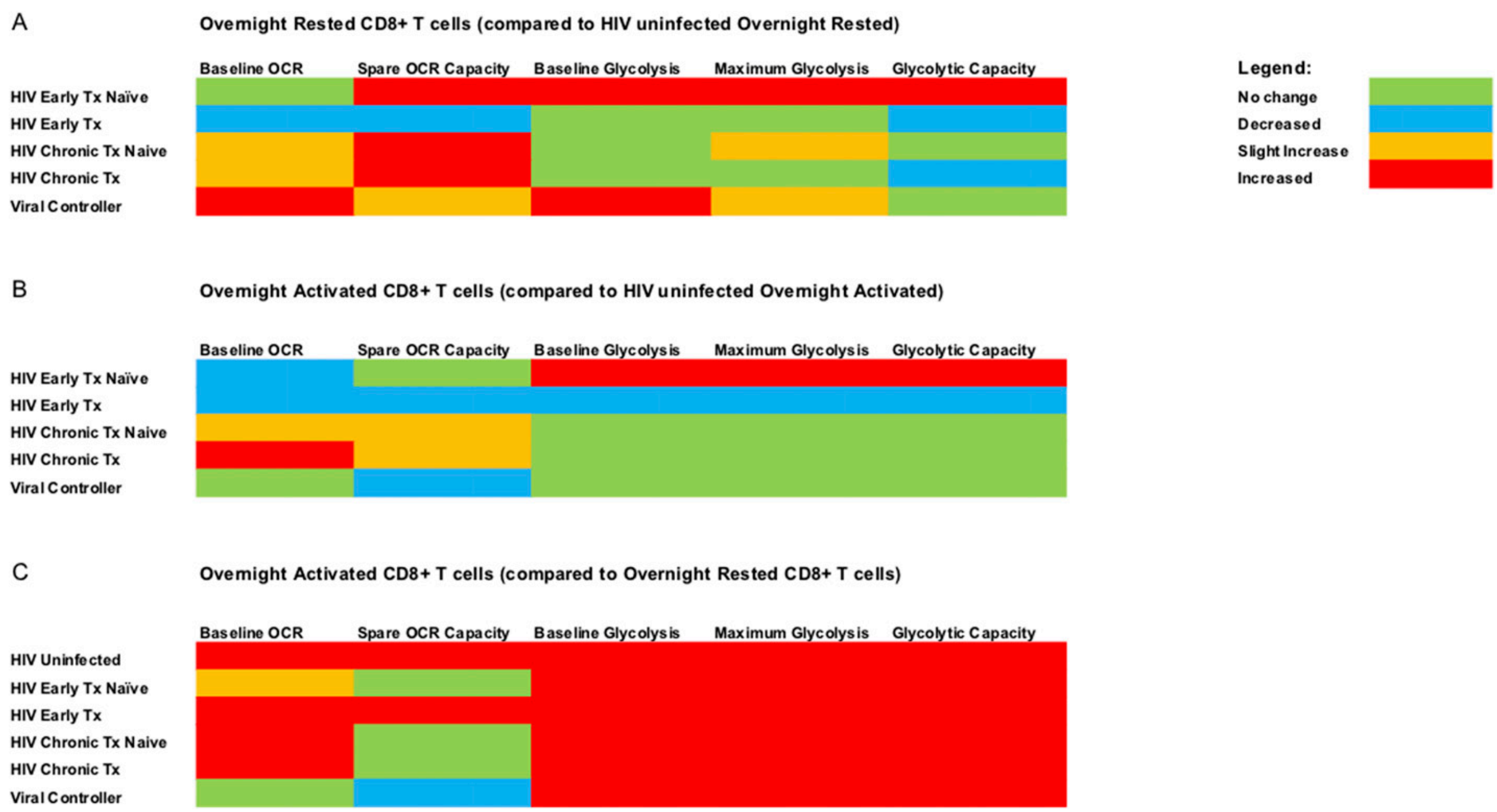


**Figure 4. Summary of metabolic profile data. A color-coded schematic diagram is provided to summarize the metabolic data generated with Seahorse extracellular flux analyzer.**
**(A)** summarizes the data of overnight rested (unstimulated) $CD8^+$ T cells from all the different HIV cohorts compared with overnight-rested $CD8^+$ T cells from HIV-uninfected donors. **(B)** summarizes the data of overnight anti-CD3/28 stimulated $CD8^+$ T cells from all the different HIV cohorts against overnight anti-CD3/28–stimulated $CD8^+$ T cells from HIV-uninfected donors. **(C)** summarizes the data of overnight-activated $CD8^+$ T cells from all the different cohorts against overnight-rested autologous $CD8^+$ T cells. The legend for the color code is on the top right corner of the figure.

was able to improve the quantity and quality of vaccine-specific responses from $CD8^+$ T cells (Araki et al, 2009; Pearce et al, 2009).

An in vitro mTORC1 inhibitory study was set up with freshly isolated PBMCs from chronic HIV-infected cART–naïve and cART-treated donors. We included two arms to this study: in one arm, PBMCs were pretreated with rapamycin for an hour, washed, and followed up with an antigen-induced 6-d proliferation assay, and in the other arm, rapamycin was included at various concentrations during the antigen-induced 6-d proliferation assay (Fig 6A). The PBMCs were stained with CFSE and stimulated with HIV Gag, HIV Nef, CMV pp65 peptide pools, or staphyloccal enterotoxin B (SEB). After 6 d, PBMCs were washed and re-stimulated overnight with peptides in the presence of Brefeldin A (BFA) to capture antigen-specific cytokines, IFN-$\gamma$ and TNF-$\alpha$ (Fig 6A). Our goal was to improve the functional outcome of antigenic responses of $CD8^+$ T cells as measured by proliferation plus cytokine production. Fig 6B shows a representative experiment from an HIV chronic cART–naïve donor. Overall, we did not observe any improvement in the proliferative and cytokine functions of $CD8^+$ T responses to HIV Gag or Nef with either transient exposure to 50 nM of rapamycin for an hour (Fig 6C and E) or titrating rapamycin at 12.5, 25, and 50 nM concentrations during the 6-d assay (Fig 6D and F). The $CD8^+$ T cells from chronic HIV-infected cART–treated donors failed to functionally respond to either HIV Gag or Nef (Fig 6E and F) in the absence of rapamycin and no improvement was observed with either rapamycin pretreatment or titration over the 6-d stimulation assay (Fig 6E and F). In all cases, having rapamycin in the system, transiently or titrated during a 6-d assay, blunted the functional response from $CD8^+$ T cells, except with SEB specific responses.

## Glycolysis is the major driver of ex vivo $CD8^+$ T-cell function

The ultimate function of an Ag specific $CD8^+$ T cell is to kill its target. We and others have shown that HIV-specific exhausted $CD8^+$ T cells are primed for killing their target but because of their functionally exhausted nature, they fail to degranulate and release perforin and granzyme at the target site (Sakhdari et al, 2012). We used $CD8^+$ T cells from chronically HIV-infected donors and manipulated metabolism with different metabolic inhibitors to investigate if the killing of HIV-infected $CD4^+$ T-cell targets by autologous $CD8^+$ T cells could be enhanced. To alter $CD8^+$ metabolism, we used 2-deoxy-D-glucose (2-DG), dichloroacetate (DCA), and oligomycin. 2-DG is a glucose analogue that is transported into cells via glucose transport molecules and binds with higher affinity to hexokinase to block glycolysis and make the mitochondria the sole source energy provider. DCA binds to pyruvate dehydrogenase kinase (PDK1) and inhibits its function to deactivate pyruvate dehydrogenase complex (PDC). PDC converts pyruvate, an end product of glycolysis, to acetyl-CoA, thus providing substrates for OXPHOS. Inhibiting PDK1 with DCA should increase enzymatic activity of PDC resulting in

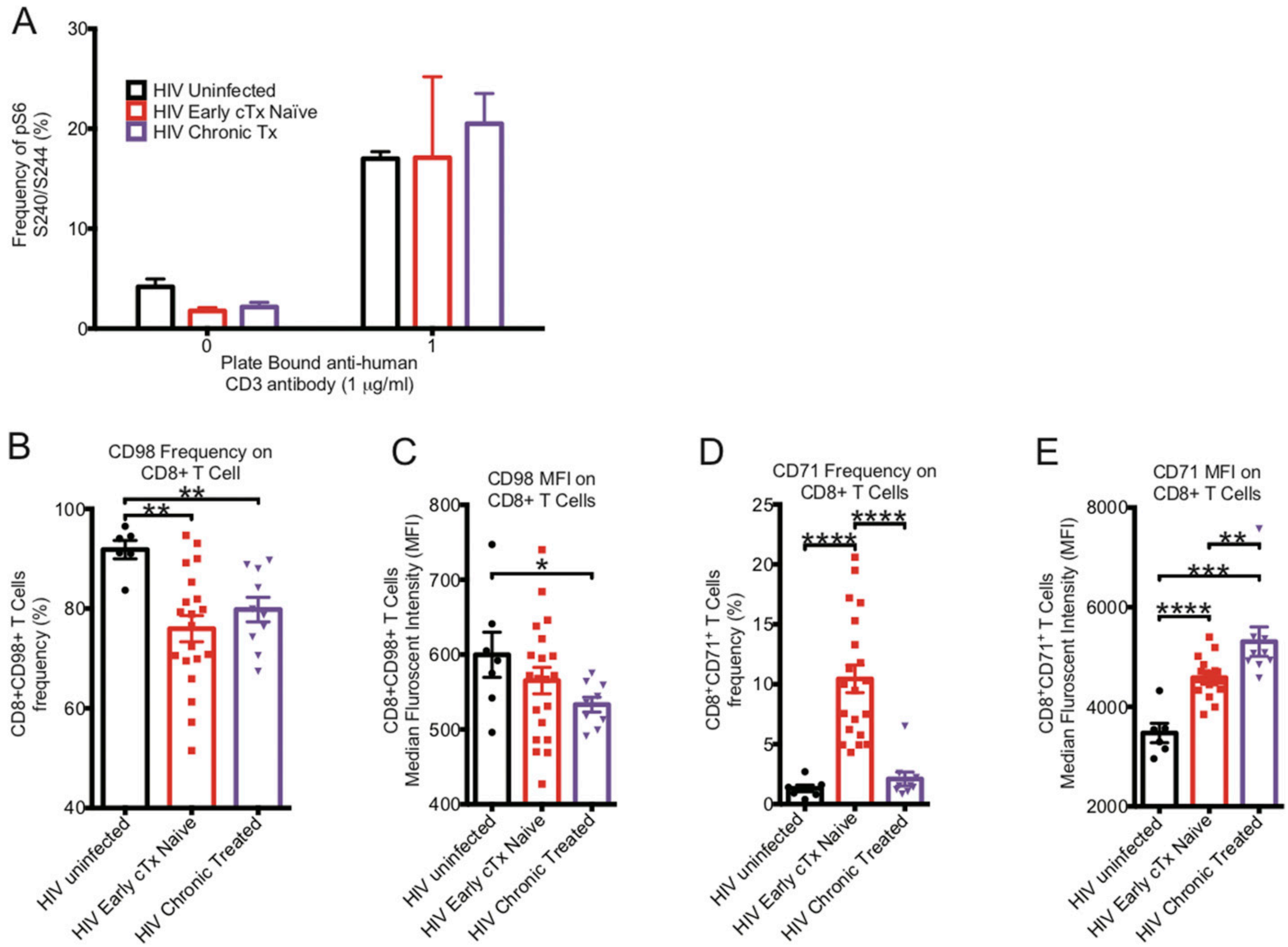


**Figure 5. mTORC1 signaling and surface transport channels in CD8+ T cells.**
**(A)** The basal mTORC1 activity in CD8+ T cells after 5 h of rest or activation with plate bound anti-human CD3 and anti-human CD28 was determined by PhosFlow in HIV-uninfected (black bar, n = 3), HIV-early cART–naïve (red bar, n = 3), and HIV-chronic cART–treated (purple bar, n = 4) donors by determining the frequency of pS6 in CD8+ T cells. **(B)** represents the frequency of CD98 expression on the surface of HIV-uninfected (black bar, n = 7), HIV-early cART (cTx)–naïve (red bar, n = 20), and HIV-chronic cART–treated (purple bar, n = 10) CD8+ T cells. **(C)** represents the median fluorescent index of CD98 expression on the surface of HIV-uninfected (black bar, n = 7), HIV-early cART–naïve (red bar, n = 20), and HIV-chronic cART–treated (purple bar, n = 10) CD8+ T cells. **(D)** represents the frequency of CD71 expression of the surface of HIV-uninfected (black bar, n = 7), HIV-early cART–naïve (red bar, n = 20), and HIV-chronic cART–treated (purple bar, n = 10) CD8+ T cells. **(E)** represents the median fluorescent index of CD71 expression on the surface of HIV-uninfected (black bar, n = 7), HIV-early cART–naïve (red bar, n = 20), and HIV-chronic cART–treated (purple bar, n = 10) CD8+ T cells. * indicates $P < 0.05$, ** $P < 0.01$, *** $P < 0.001$, and **** $P < 0.0001$ by unpaired two-tailed nonparametric Mann–Whitney test. Error bars are mean ± SEM.

excess pyruvate to be shunted into mitochondria as acetyl-CoA to increase OXPHOS and reduce glycolysis (Ruggieri et al, 2015; Shen et al, 2015). Oligomycin is an antibiotic that inhibits ATP synthase by blocking proton channels and concurrently enhances glycolysis.

To test the effect of either blocking glycolysis or OXPHOS on HIV-specific mediated killing we performed killing assays using PBMCs from chronically HIV-infected individuals, in which autologous CD8+ T cells were co-cultured with activated CD4+ T cells that were infected with exogenous HIV (Fig 7A). Exogenous HIV infection of CD4+ T cells was performed to ensure a high frequency of infected cells to be recognized by autologous CD8+ T cells. The night before setting up the co-culture killing assay, autologous CD8+ T cells were negatively isolated and rested overnight in media. On the day of the killing assay, exogenously HIV-infected CD4+ T cells were co-cultured with overnight-rested autologous CD8+ T cells and then assessed the following day for killing by measuring the frequency of p24-expressing CD4+ T cells that remained in the culture compared to the wells that were seeded with similar number of exogenous HIV-infected CD4+ T cells without CD8+ T cells to determine the maximum frequency of HIV-infected CD4+ T cells (see the Materials and Methods section). As a control for maximal CD8+ T-cell killing activity, an aliquot of CD8+ T cells were activated overnight with plate-bound anti-human CD3 and anti-human CD28 antibodies to enhance the baseline glycolysis and OXPHOS rates of the CD8+ T cells. To compare the effects of the metabolic inhibitors on killing, inhibitors, 2-DG, DCA, and oligomycin, were individually added during the co-culture killing assay (Fig 7C–G). In all killing assays, CD8+ T cells were added to HIV-infected CD4+ T cells targets at a 1:1

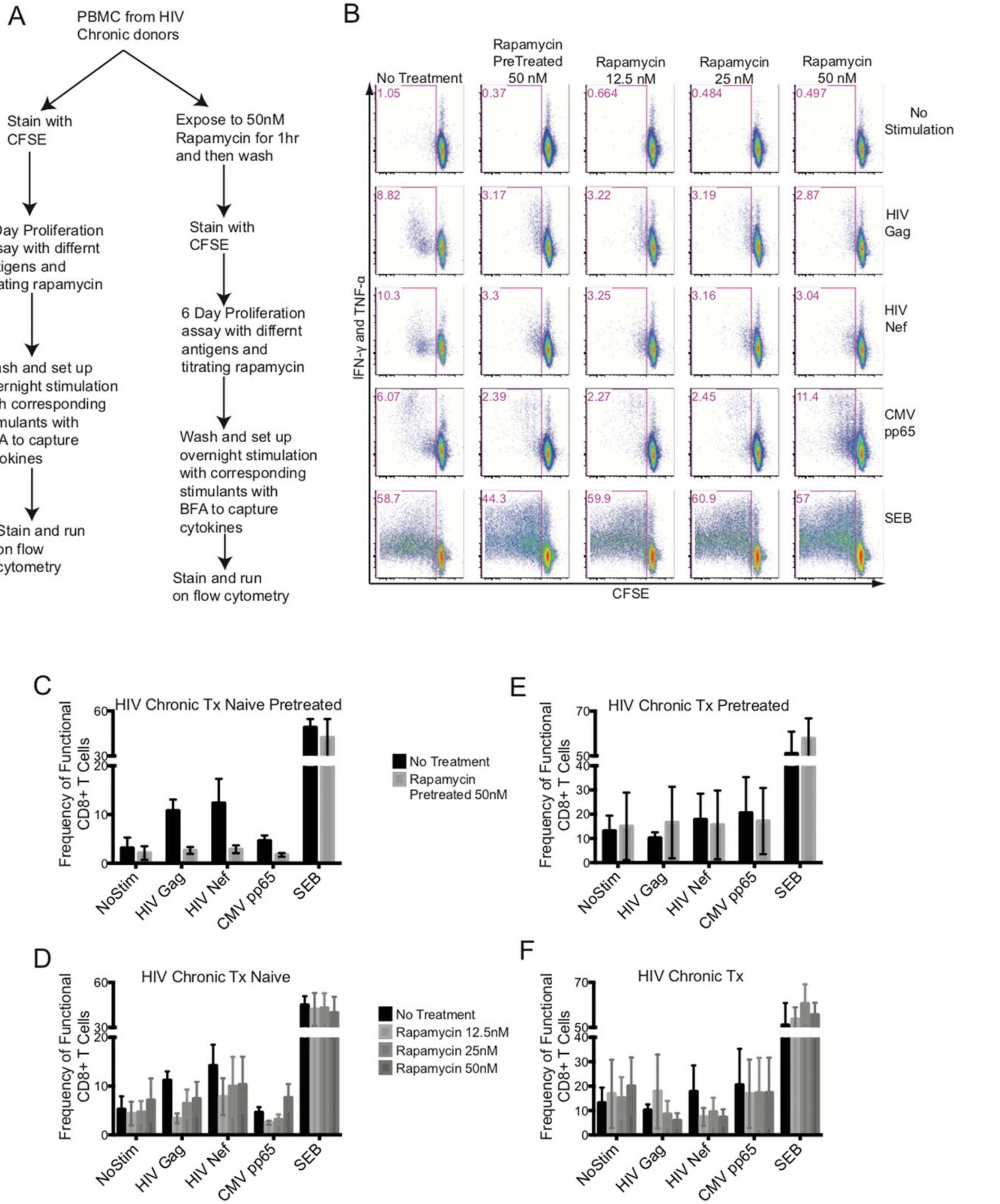


**Figure 6. mTORC1 inhibition does not rescue exhausted HIV-specific CD8+ T-cell function.**

**(A)** flowchart of the experimental design. **(B)** illustrates a representative data set from an HIV-chronic cART–naïve donor. The CD8+ T-cell function was plotted with CFSE on the x-axis (proliferation) against intracellular IFN-γ plus TNF-α on the y-axis (response to antigen or mitogen). Columns represent rapamycin treatment and rows represent stimulation conditions. **(C)** represents the antigen specific functional output of CD8+ T cells from HIV-chronic cART–naïve donors pretreated with 50 nM of rapamycin. **(D)** represents the antigen-specific functional output of CD8+ T cells from HIV-chronic cART–naïve donors treated with 12.5, 25, and 50 nM of rapamycin during the 6 d culture. **(E)** represents the antigen specific functional output of CD8+ T cells from HIV-chronic cART–treated donors pretreated with 50 nM of rapamycin. **(F)** represents the antigen-specific functional output of CD8+ T cells from HIV-chronic cART–treated donors treated with 12.5, 25, and 50 nM of rapamycin during the 6 d culture. HIV-chronic cART–naïve n = 6, and HIV-chronic cART–treated n = 6. Error bars are mean ± SEM.

ratio. In some experiments, we also tested if overnight rested CD8+ T cells that were pretreated with oligomycin (Fig 7F) and DCA (Fig 7G) for 3 h and then washed had a different effect on killing than overnight exposure to metabolic inhibitors during the killing assay. Overnight-rested CD8+ T cells were able to decrease the frequency of exogenously HIV-infected CD4+ T cells by 40.7% ± 6.0% on average (Fig 7B). Overnight activated CD8+ T cells were able to eliminate their HIV-infected CD4+ T cells targets at a significantly higher rate of 53.1% ± 6.2% (**P* = 0.0137, Fig 7B). Adding 2-DG and DCA during the killing assay suppressed the killing ability of the overnight rested CD8+ T cells by 33.9% ± 2.6% and 22.3% ± 6.3%, **P* = 0.0156, respectively (Fig 7C and D). Oligomycin did not have an overall impact on the killing ability (40.7% ± 4.2%) of the CD8+ T cells, although in CD8+ T cells from three individuals, we did see enhancement (Fig 7E). Transiently exposing CD8+ T cells to oligomycin and DCA had a slight increase (49.0% ± 6.5%) and similar (40.0% ± 7.6%) killing ability of the autologous CD8+ T cells, respectively (Fig 7F and G). These results indicate that blocking glycolysis reduced killing of the target cells by CD8+ T cells. Enhancing OXPHOS by blocking glycolysis with 2-DG and DCA had an overall negative effect on the killing ability of the CD8+ T cells (Fig 7C and D). However, enhancing glycolysis with oligomycin by blocking OXPHOS (Fig 7E), even transiently (Fig 7F), was able to maintain or increase baseline killing ability of the CD8+ T cells, indicating that glycolysis had a more important role than OXPHOS on the functional target elimination abilities of the CD8+ T cells.

### Cytoplasmic glyceraldehyde-3-phosphate dehydrogenase (cGAPDH) in CD8+ T cells

GAPDH is a 37-kD protein that is found in different subcellular regions within a cell with most detected in the cytoplasm and membrane regions. The presence of cytoplasmic GAPDH (cGAPDH) in a cell is associated with higher glycolytic activity and with enhanced glycolytic capacity (Mazzola & Sirover, 2003). In activated CD8+ T cells, GAPDH translocates from the nucleus to the cytoplasm and elevated amounts of cGAPDH are directly correlative of CD8+ T-cell function (Chang et al, 2013; Gubser et al, 2013; Dimeloe et al, 2016). We investigated the cytoplasmic localization of GAPDH in CD8+ T cells from HIV-uninfected, early HIV-infected cART–naïve, early HIV-infected cART–treated, chronic HIV-infected cART–treated, and HIV VCs donors with CMV- and HIV-specific dextramers using an ImageStream imaging flow cytometer. We used similarity feature to determine the frequency of CD8+ T cells with cGAPDH (Fig S4A–F). In HIV-uninfected donors, GAPDH was present in the cytoplasm of 77.7% ± 2.7% of CMV nonspecific (CMV dextramer−) and in 73.8% ± 1.4% of CMV-specific (CMV dextramer+) CD8+ T-cell population (Fig 8A). Thus, cGAPDH is present in the CMV nonspecific and CMV specific-CD8+ T cells in the HIV-uninfected donors. In ex vivo CD8+ T cells from HIV-infected individuals, we see a systematic reduction in the frequency of CD8+ T cells with cGAPDH within and outside dextramer-positive populations compared with HIV-uninfected (Fig 8A–E), with the least reduction in VCs. cGAPDH MFI, however, was preserved in positive cells or and slightly elevated in early infection (Fig 8F–J). Despite systemic reduction of CD8+ T cells with cGAPDH in the HIV group, HIV dextramer–positive cells tended to show greater frequency of CD8+ T cells with cGAPDH frequencies in the early cART–naïve group and the VC group (Fig 8K–R). Of note was that HIV dextramer–positive cells from early HIV-infected cART–naïve displayed the greatest cGAPDH MFI of all samples studied (Fig 8). In summary, we find a systematic decrease in the frequency of cGAPDH+ CD8+ T cells in HIV–infected individuals, which is generally more reduced in antigen specific CD8+ T cells (CMV or HIV), and this defect was not restored with treatment. Chronic HIV-infected cART–treated donors had the least amount of cGAPDH content in CD8+ T cells based on MFI. The one exception we observed was with HIV-specific CD8+ T cells during early HIV-infected cART–naïve donors, which had more cGAPDH containing CD8+ T cells both in frequency and MFI compared with other cell populations. Although HIV VCs had more of cGAPDH CD8+ T cell in terms of frequencies compared with other HIV stages, their cGAPDH content was not increased. In summary, cGAPDH was reduced in all CD8+ T cells; however, the reduction was least in early HIV-infected cART–naïve donors, which also correlates with greatest glycolytic capacity.

## Discussion

In this work, we investigate functional metabolic pathophysiological processes in CD8+ T cells within the context of HIV infection. HIV is not curable and requires lifelong cART. Any cessation of cART leads to rebound viremia because of a persistently infected reservoir (Kim et al, 2014; Hosmane et al, 2017). The distinct clinical phases of HIV infection allowed us to investigate the metabolic status of CD8+ T cells in various stages of virus control. During acute/early HIV infection, one observes partial control of virus, which then stabilizes toward a "set-point" viral level because of multiple host and viral factors leading to chronic persistent viremia, the latter associated with marked CD8+ T-cell exhaustion (Kahn & Walker, 1998). Thus, we predicted that the metabolic profiles of ex vivo CD8+ T cells would be different in HIV versus uninfected states. In this regard, in contrast to HIV-uninfected individuals, we found that ex vivo CD8+ T cells from early HIV-infected cART–naïve, chronic HIV-infected cART–naïve, chronic HIV-infected cART–treated and HIV VCs operated at maximal OXPHOS capacity even at rest because subsequent anti-human CD3 and anti-human CD28 stimulation did not enhance spare OCAR capacity of these cells (see Fig 4 summary diagram). This indicates that mitochondrial energy demands are high during early untreated and chronic infection, as well as in states of excellent viral control. CD8+ T cells, however, from HIV early cART–treated individuals had reduced OXPHOS capacity even compared to HIV-uninfected individuals, but their cells were able to regain spare OXPHOS capacity after stimulation but not to the level of uninfected groups. We assessed the relative effects of glycolysis and OXPHOS on functional killing by CD8+ T cells. Our data indicate that glycolysis was the predominant metabolic pathway for proliferative, cytokine, and killing abilities of CD8+ T cells, although OXPHOS also may have contributed for maximal functional activity. CD8+ T cells from early HIV-infected cART–naïve and HIV VCs showed elevated levels of baseline glycolysis, the primary driver of functionality in these CD8+ T cells. This is consistent with the observation that viral control by CD8+ T cells is most apparent in these clinical stages of HIV infection. mTORC1 activation has been linked with enhanced glycolysis. Surprisingly,

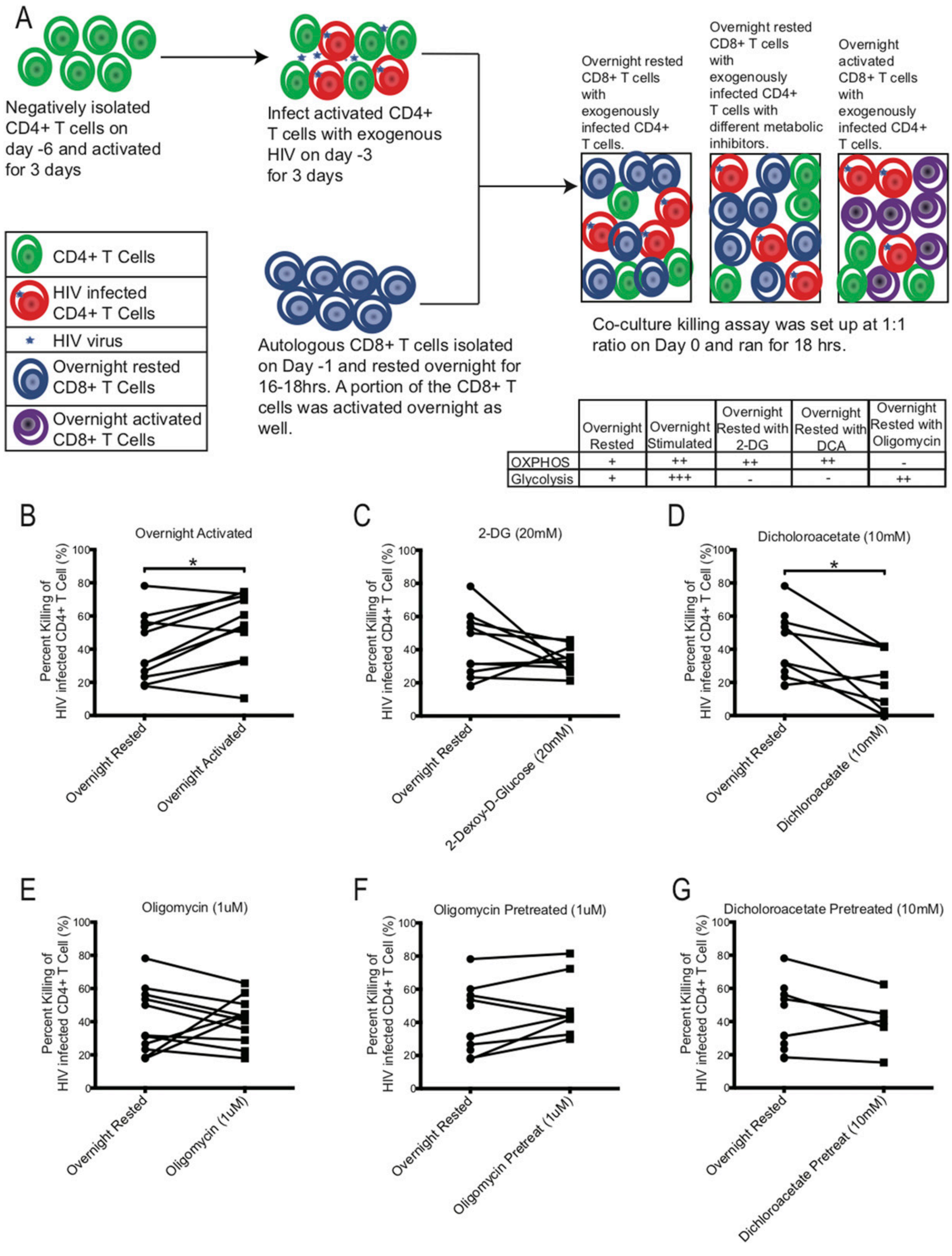


**Figure 7. Effect of metabolic inhibitors on HIV-specific killing capacity of CD8⁺ T cells.**
**(A)** Schematic of killing assay experimental setup. Table indicates proposed effect of inhibitors on metabolism. **(B, C, D, E, F, G)** The killing capacity of $CD8^+$ T cells from overnight rested (n = 11) donors were compared against $CD8^+$ T cells that were either: (B) activated overnight (n = 11) or (C) had 20 mM of 2-DG (n = 10) or (D) 10 mM of DCA (n = 8) or (E) 1 μM of oligomycin (n = 11) during killing assay, or (F) were pretreated with 1 μM of oligomycin (n = 8) or (G) 10 mM of DCA (n = 5) and washed before setting up the killing assay. * indicates $P < 0.05$ by unpaired two tailed nonparametric Mann–Whitney test.

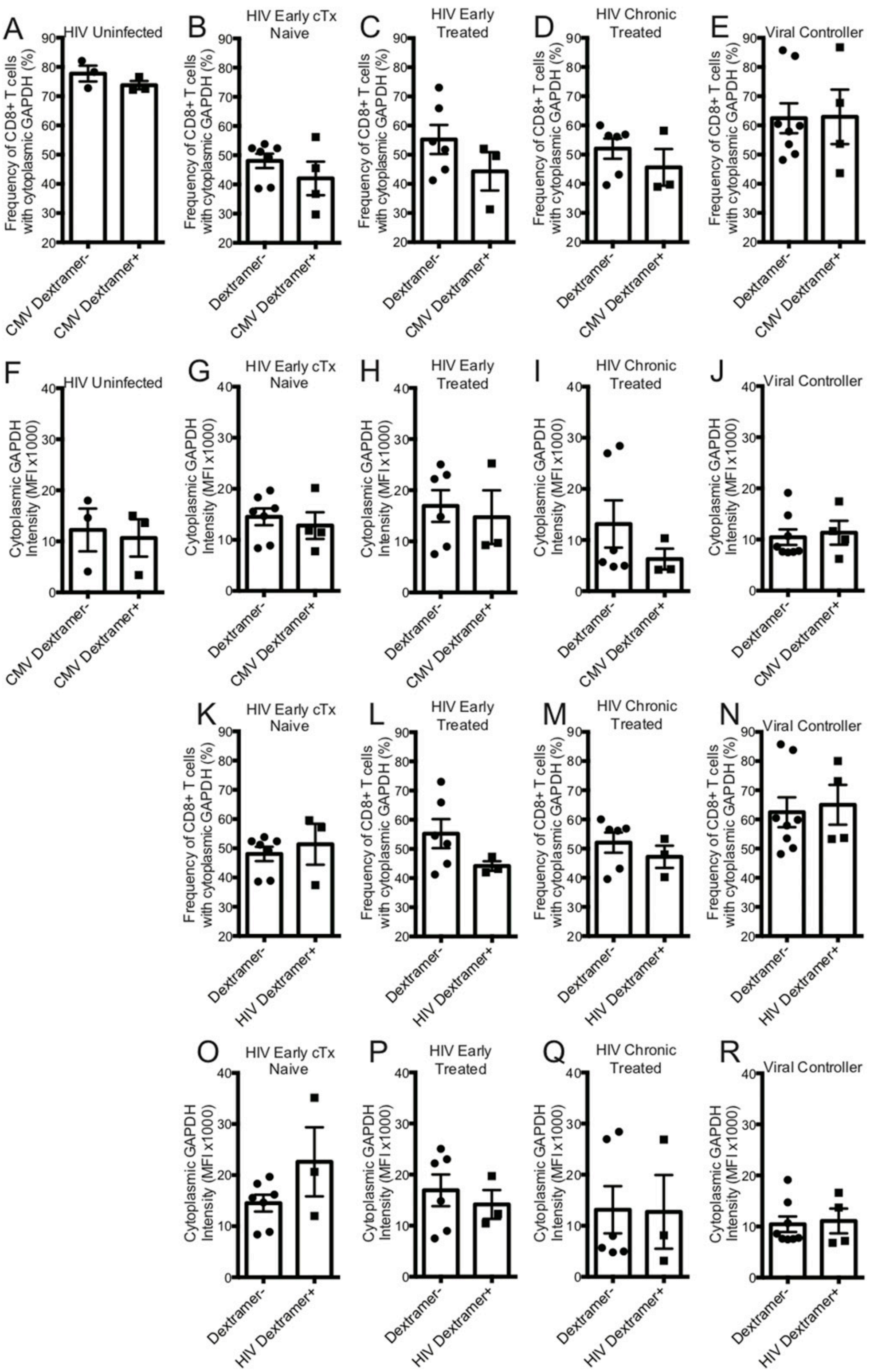


**Figure 8. GAPDH localization in ex vivo CD8⁺ T cells from HIV-uninfected and HIV-infected individuals.**

**(A)** represents the frequency of CMV nonspecific (dextramer−) and CMV-specific (dextramer+) CD8⁺ T cells with cGAPDH from HIV-uninfected (n = 3) donors. **(B)** represents the frequency of dextramer− and CMV specific (dextramer+) CD8⁺ T cells with cGAPDH from HIV-early cART–naïve (n = 4) donors. **(C)** represents the frequency of dextramer− and CMV-specific (dextramer+) CD8⁺ T cells with cGAPDH from HIV-early cART–treated (n = 3) donors. **(D)** represents the frequency of dextramer− and CMV-specific (dextramer+) CD8⁺ T cells with cGAPDH from HIV-chronic cART–treated (n = 3) donors. **(E)** represents the frequency of dextramer− and CMV-specific (dextramer+) CD8⁺ T cells with cGAPDH from viral controller (VC) (n = 3) donors. **(F)** represents the median fluorescent index (MFI) of cytoplasmic GAPDH intensity of CMV nonspecific (dextramer−) and CMV-specific (dextramer+) CD8⁺ T cells from HIV-uninfected (n = 3) donors. **(G)** represents the MFI of cytoplasmic GAPDH intensity of dextramer− and CMV-specific (dextramer+) CD8⁺ T cells from HIV-early cART–naïve (n = 4) donors. **(H)** represents the MFI of cytoplasmic GAPDH intensity of dextramer− and CMV-specific (dextramer+) CD8⁺ T cells from HIV-early cART–treated (n = 3) donors. **(I)** represents the MFI of cytoplasmic GAPDH intensity of dextramer− and CMV-specific (dextramer+) CD8⁺ T cells from HIV-chronic cART–treated (n = 3) donors. **(J)** represents the MFI of cytoplasmic GAPDH intensity of dextramer− and CMV-specific (dextramer+) CD8⁺ T cells from VC (n = 4) donors. **(K)** represents the frequency of dextramer− and HIV-specific (dextramer+) CD8⁺ T cells with cGAPDH from HIV-early cART–naïve (n = 3) donors. **(L)** represents the frequency of dextramer− and HIV-specific (dextramer+) CD8⁺ T cells with cGAPDH from HIV-early cART–treated (n = 3) donors. **(M)** represents the frequency of dextramer− and CMV-specific (dextramer+) CD8⁺ T cells with cGAPDH from HIV-chronic cART–treated (n = 3) donors. **(N)** represents the frequency of dextramer− and HIV-specific (dextramer+) CD8⁺ T cells with cGAPDH from VC (n = 4) donors. **(O)** represents the MFI of cytoplasmic GAPDH intensity of dextramer− and HIV-specific (dextramer+) CD8⁺ T cells from HIV-early cART–naïve (n = 3) donors. **(P)** represents the MFI of cytoplasmic GAPDH intensity of dextramer− and HIV-specific (dextramer+) CD8⁺ T cells from HIV-early cART–treated (n = 3) donors. **(Q)** represents the MFI of cytoplasmic GAPDH intensity of dextramer− and HIV-specific (dextramer+) CD8⁺ T cells from HIV-chronic cART–treated (n = 3) donors. **(R)** represents the MFI of cytoplasmic GAPDH intensity of dextramer− and HIV-specific (dextramer+) CD8⁺ T cells from VC (n = 4) donors. The dextramer− population was combined within each of the different HIV-infected cohorts for analysis purposes. Error bars are mean ± SEM.

we find that the CD8$^+$ T cells had reduced mTORC1 activity in early and chronic HIV infection, despite high levels of immune activation indicated by the surface expression of CD71. This is consistent with our findings of reduced baseline glycolysis and glycolytic capacity in chronic HIV. During chronic HIV infection, baseline glycolysis levels were similar to those from HIV-uninfected donors. Glycolysis, however, was enhanced during early HIV-infected cART–naïve CD8$^+$ T cells despite low mTORC1 activity. We are unable to completely understand this observation; however, we postulate that if mTORC1 activity were enhanced, glycolytic capacity could be enhanced further suggesting a relative defect of glycolysis during acute infection. It is unclear whether these cells had prior mTORC1 activation which then was down-regulated with persistence of a glycolytic program. Further kinetics of mTORC1 activation during early infection would need to be performed to determine this. Our attempts to improve proliferative and cytokine functions in peptide antigen-stimulated CD8$^+$ T cell from chronic HIV-infected donors in the absence or presence of rapamycin, an mTORC1 inhibitor, were unsuccessful. In addition, despite previous works in murine systems, mTORC1 inhibition could not reverse CTL defects from HIV-infected individuals (D'Souza et al, 2011). These latter findings also support our observation of reduced, rather than excessive mTORC1 activity in ex vivo CD8$^+$ T cells in HIV. The etiology of this reduced mTORC1 activation in the face of ongoing viral replication is unclear.

cGAPDH is associated with enhanced glycolytic ability/capacity. We found that HIV infection induces a systemic depression in cGAPDH containing CD8$^+$ T cells; however, this deficiency was at least countered in HIV-specific CD8$^+$ T cells during early HIV-infected cART–naïve stage by having higher contents of cGAPDH and in HIV VCs by greater frequency of cGAPDH containing cells, suggesting some preservation of CD8$^+$ T-cell function by enhancing glycolytic capacity during these clinical states. Thus, our results indicate that glycolytic activity and capacity of CD8$^+$ T cells associate with better HIV control.

CD8$^+$ T cells from early HIV-infected cART–naïve donors, where partial control of viremia is well documented, and from HIV VCs, who can control HIV viremia without cART for a long time, had significantly elevated baseline glycolysis rates. Because the CD8$^+$ T cells of HIV VCs are able to control HIV replication, they appear to maintain their functional capabilities with enhanced glycolysis and maximal OXPHOS activity. The CD8$^+$ T cells from early HIV-infected cART–naïve donors were using spare capacity OCR at maximal rate after overnight rest as well. The maximized use of spare capacity OCR suggests that the CD8$^+$ T cells from early HIV-infected cART–naïve and HIV VCs donors, including those from chronic HIV-infected cART–naïve and chronic HIV-infected treated donors, had limited ability to generate additional energy through oxygen consumption in situations of metabolic stress, yet could up-regulate glycolysis to meet such energetic demands. During chronic HIV infection, it is well established that CD8$^+$ T cells are rendered functionally exhausted (Day et al, 2006; Jones et al, 2008; Sakhdari et al, 2012; Buggert et al, 2014; Wolski et al, 2017). Fig 9A illustrates that chronic HIV-infected cART–naïve donors for our metabolic extracellular flux experiments had elevated levels of PD-1 expression on CD8$^+$ T cells, indicating their exhausted status, but with cART treatment, the frequency of CD8+PD-1+ T cells decreased to that observed in HIV-uninfected donors. Once exhausted, they lose their ability to derive energy from elevated levels of glycolysis, and reduce baseline glycolysis rates to that observed in HIV-uninfected donors while maintaining slightly elevated OXPHOS rates with maximizing utilization of spare capacity OCR. Similar to their counterparts from early HIV-infected cART–naïve and HIV VCs, CD8$^+$ T cells from chronic HIV-infected donors also had reduced capability to derive additional energy from OXPHOS after TCR re-stimulation, and with the loss of elevated glycolysis, we postulate that they become functionally exhausted. It is unclear, however, from our data, whether functional exhaustion was due to low glycolytic activity or whether glycolytic activity reflected functional exhaustion due to inhibition of T-cell signaling pathways at the TCR, such as PD-1 or other checkpoint inhibitor engagement. We likely favour the later, due to the fact that the CD8$^+$ T cells from all the different cohorts could tap into their glycolytic capacity ECAR for additional energy after overnight anti-human CD3 and anti-human CD28 co-stimulation. The etiology of CD8$^+$ T-cell exhaustion in HIV is multifactorial. The constant presence of HIV Ag and the gradual decline of CD4$^+$ T cells, among other reasons, exacerbate the functional exhaustion status of CD8$^+$ T cells (Wherry et al, 2003). In addition, we have identified that the CD8$^+$ T cells from the chronic HIV phase had "dysregulated" baseline glycolysis as the rates were similar to that of HIV-uninfected donors, rendering them unable to functionally exert control of HIV infection with elevated levels of glycolysis, and thus remain functionally exhausted. The metabolic functional profiles of CD8$^+$ T cells from early HIV-infected cART–treated were unusual. The reduced OXPHOS and glycolytic capacity in these cells even compared to uninfected samples suggests a metabolically suppressed state, which will require further investigation.

The metabolic nature of exhaustion of CD8$^+$ T cells in HIV infection is different from other disease states. Exhausted hepatitis B virus (HBV)–specific CD8$^+$ T cells surprisingly are able to generate energy from glycolysis but have impaired capacity to use mitochondrial energy (OXPHOS), indicating that exhaustion in these cells are primarily driven by defective mitochondrial function (Schurich et al, 2016). These CD8$^+$ T cells are unable to maximize their energy generation by coupling glycolysis with OXPHOS. In a tumor microenvironment, among other causes, exhaustion of CD4$^+$ T-cell function is imposed by nutrient depletion, which is reversible in nutrient replete environments (Chang et al, 2013).

Similar to other studies, we have identified that CD8$^+$ T cells function to maximize energy generation by combining glycolysis with OXPHOS (Schurich et al, 2016). To gain further insight into the relative usage of glycolysis and OXPHOS, we plotted OCR versus ECAR at baseline from overnight rested and overnight activated conditions. Fig 9B illustrates that the CD8$^+$ T cells were primarily deriving their energy by OXPHOS after overnight rest. It is interesting to note that the CD8$^+$ T cells from chronic HIV-infected cART–naïve and cART–treated donors clustered with CD8$^+$ T cells from HIV-uninfected donors. The CD8$^+$ T cells from HIV VCs and early HIV-infected cART–naïve were further out towards the dotted boundary line, being on top of which indicates equal energy generation from glycolysis and OXPHOS. After overnight anti-human CD3 and anti-human CD28 activation, the CD8$^+$ T cells from all donors, except from HIV VCs, increased energy generation by up-regulating both glycolysis and OXPHOS (Fig 9C). CD8$^+$ T cells from early HIV-infected cART–naïve were generating almost half of their energy through glycolysis. The CD8$^+$ T cells from early

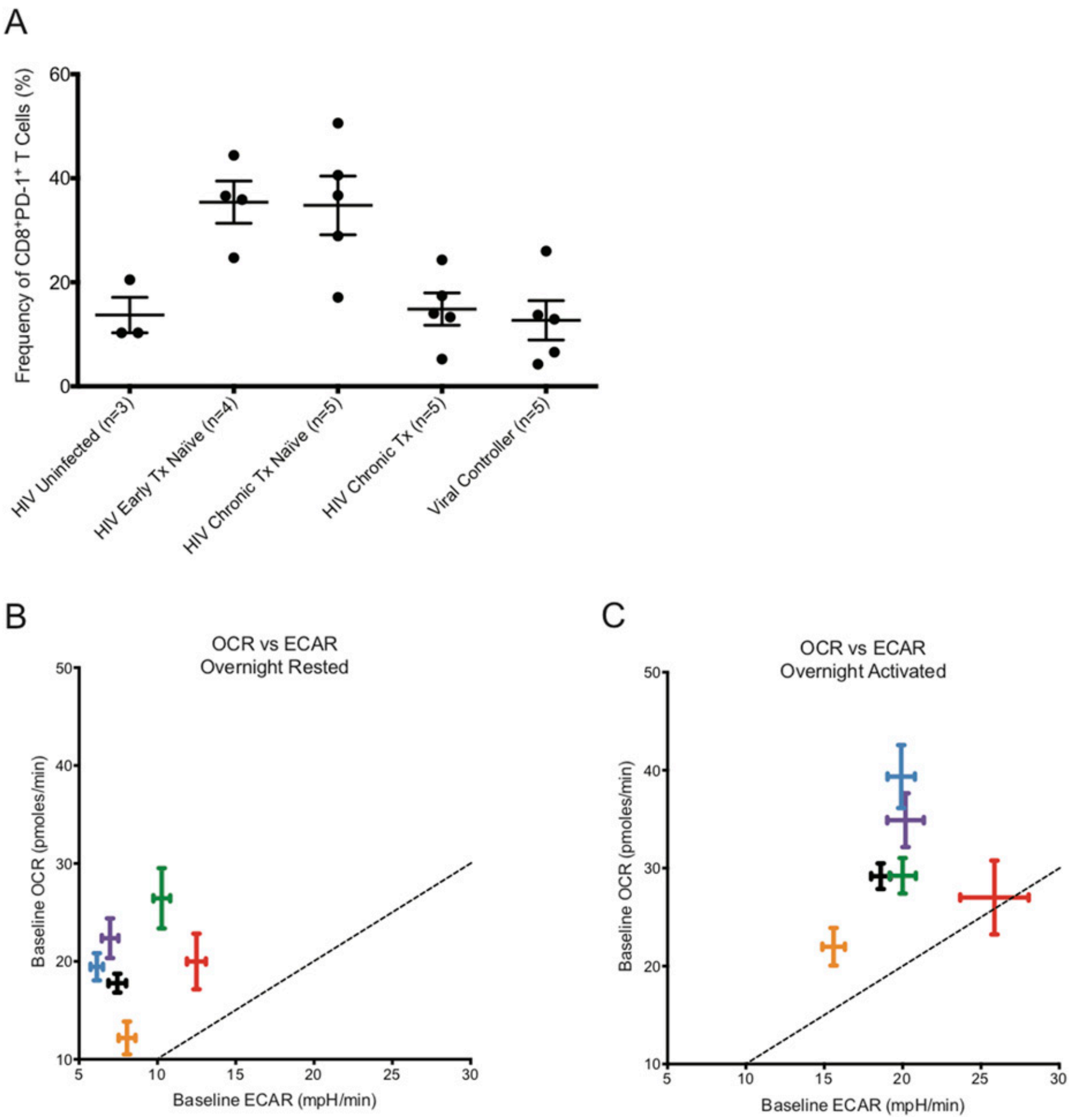


**Figure 9. PD-1 expression frequency on CD8⁺ T cells with oxygen consumption rate (OCR) versus ECAR plot for CD8⁺ T cells taken from HIV-uninfected and HIV-infected individuals.**
**(A)** represents the frequency of PD-1 expression on the surface of CD8⁺ T cells from HIV-uninfected (n = 3), HIV-early cART–naïve (n = 4), HIV-chronic cART–naïve (n = 5), HIV-chronic cART–treated (n = 5), and viral controllers (VCs) (n = 5). **(B)** represents the baseline OCR versus baseline ECAR plot of HIV-uninfected (black), HIV-early cART–naïve (red), HIV-early cART–treated (orange), HIV-chronic cART–naïve (blue), HIV-chronic cART–treated (purple), and VCs (green) of ex vivo overnight-resting (unstimulated) CD8⁺ T cells. **(C)** represents the baseline OCR versus baseline ECAR plot of HIV-uninfected (black), HIV-early cART–naïve (red), HIV-early cART–treated (orange), HIV-chronic cART–naïve (blue), HIV-chronic cART–treated (purple), and VCs (green) after overnight CD3/28 activation of ex vivo CD8⁺ T cells. These data were compiled from the data set from Figs 1–3. Dotted lines represent the points on the graph where ECAR = OCR.

HIV-infected cART–naïve have been programed to balance the tradeoff between exhausting local resources by rapid consumption to sustain ATP production against the survival of the host (Pfeiffer et al, 2001). Hence, we postulate that the elevated levels of glycolysis observed in CD8⁺ T cells from early HIV-infected cART–naïve donors are not sustainable over the long term, and thus during the chronic phase of HIV infection, the metabolic profiles of CD8⁺ T cells begin to resemble their counterparts from HIV-uninfected donors in terms of relative energy generation from glycolysis and OXPHOS, despite the presence of virus. HIV VCs constitute less than 1% of the total cases of HIV infection. Their CD8⁺ T cells are unique in that they can continue generating energy from elevated levels of glycolysis for a longer term to control HIV viremia. CD8⁺ T cells from VCs at rest had the highest baseline OXPHOS activity and relative enhanced glycolysis, consistent with demands of maintaining a memory pool as well as effector function to control virus. These latter findings are consistent with the metabolic studies of Angin et al (2019) who demonstrate that CD8⁺ T cells from VC have the greatest metabolic plasticity to function in a way that they can rely on both glycolysis and oxidative phosphorylation to function as potent effector cells (Angin et al, 2019).

We were unable to recuperate the functionality of CD8⁺ T cells from chronic HIV-infected donors by mildly inhibiting the function of dysregulated mTORC1 kinase activity with rapamycin. Although early exhausted CD8⁺ T cell in lymphocytic choriomeningitis virus infection had hyperactive mTORC1 activity (Bengsch et al, 2016), in HIV, this may not be the case (Fig 5) and our phenotypic and functional studies only confirmed this phenomenon (Figs 5 and 6). A recent study has shown that rapamycin treatment led to depressed glycolysis while maintaining OXPHOS in peptide-stimulated CD8⁺ T cells. (Hukelmann et al, 2016). Through this rapamycin inhibition study, we are thus able to indirectly establish that glycolysis is the main driver of antigen specific CD8⁺ T-cell proliferation and cytokine functions. Overall, our data suggest a relative defect in mTOR activation occurs during HIV infection and future work examining potential mTOR activators should be examined on enhancing HIV-specific CD8⁺ T-cell function. It is also possible that mTORC1 is initially activated very early in the CD8⁺ T-cell immune response which then gets rapidly suppressed with ongoing antigen stimulation through undefined mechanisms.

GAPDH is a multifunctional protein, involved not only in glycolysis but in other cellular processes, including but not limited to DNA repair, tRNA export, mRNA regulation, membrane fusion and transport, cytoskeletal dynamics, and cell death (reviewed in Nakano et al [2018] and Tristan et al [2011]). The subcellular localization of GAPDH displays a predefined temporal sequence that is dependent on the proliferative state of the cell. In proliferating cells, GAPDH localizes from cytoplasm to nucleus, and in contrast translocates from nucleus to cytoplasm in cells with diminished to

arrested proliferation (Cool & Sirover, 1989). Furthermore, recently GAPDH, was shown to be a critical agent determining cell cycle progression (Kim et al, 1999; Phadke et al, 2009, 2011). In our study, we've observed CD8+ T cells from HIV-uninfected individuals have high cGAPDH. With HIV infection, the proportion of CD8+ T cells with cGAPDH decreases by 38%, but remarkably there was a corresponding huge increase in CD8+ T cells with nuclear GAPDH (nGAPDH) (data not shown). This translocation of GAPDH from cytoplasm to nucleus reflects the proliferative burst that CD8+ T cells undergo to partially control HIV infection during early HIV-infected cART–naïve phase. To our surprise, the decrease in CD8+ T cells with cGAPDH with the concurrent increase in CD8+ T cells with nGAPDH is a systemic phenomenon within the global CD8+ T cells, not limited to only HIV specific CD8+ T cells. This altered subcellular GAPDH localization is maintained even after cART treatment during early and chronic HIV phases. Effective cART treatment, which brings the HIV viremia to undetectable levels, could not restore the cGAPDH content of CD8+ T cells, suggesting some irreversibility in this altered status quo of the cGAPDH and nGAPDH ratio in CD8+ T cells.

In conclusion, we have defined the functional metabolic profiles of CD8+ T cells in HIV infection. Elevated glycolysis levels positively correlate with partial and full control of HIV infection. Glycolysis has a bigger impact on CD8+ T-cell function, which may be driven by significantly more cGAPDH in CD8+ T cells. HIV infection is also associated with reduced mTORC1 activity likely contributing to relative reduction in glycolytic activity. The altered subcellular GAPDH localization observed in early HIV-infected cART–naïve phase is maintained even after being on treatment. In future studies, metabolic therapeutics that can enhance glycolytic capacity and mTOR1 activation should be considered along with effective cART to recuperate metabolic functional profiles of CD8+ T cells to maximize their functional control of HIV.

## Materials and Methods

### Ethics statement

Informed consent was obtained in accordance with the guidelines for conducting the clinical research at the University of Toronto and Maple Leaf Clinic institutional ethics boards. Written informed consent was provided for this study, which was reviewed by research ethics board of the University of Toronto and of St. Michael's Hospital.

### Patient groups

We have studied individuals from three different clinical phases of HIV infection, as they are likely to have important roles in immunopathogenesis: (1) early HIV infection, infected within 6 mo, (2) Chronic progressive HIV infection, HIV-infected for at least 1 yr with CD4+ T-cell count <500; and (3) VCs, asymptomatic with untreated HIV infection for at least 1 yr with no consistent decline in peripheral blood CD4 count and low or undetectable levels of plasma viremia (<5,000 copies/ml). We have prospectively matched samples of subjects before and after at least 1 yr on HAART and HIV-uninfected individuals. HIV-early Tx–naïve individuals had an average viral load of $3.6 \times 10^5$ copies/ml with average CD4+ T-cell count of 533/μl. HIV-chronic Tx–naïve individuals had an average viral load of $6.4 \times 10^4$ copies/ml with an average CD4+ T-cell count of 415/μl. VCs had an average viral load of 347 copies/ml with average CD4+ T-cell count of 864/μl.

### Metabolic analysis

Real-time changes in OCRs (as a measure of OXPHOS) and (as a measure of lactate production) were measured by extracellular flux analysis using a Seahorse XFe96 Analyzer (Agilent Technologies Inc.) at SPARC Biocentre (The Hospital for Sick Children). In brief, CD8+ T cells were negatively isolated from cryopreserved PBMCs using EasySep Human CD8+ T-cell isolation kit according to the manufacturer's protocol (STEMCELL Technologies Canada Inc.). Half of the isolated CD8+ T cells was rested overnight in R-10 media and the other half was activated overnight on plate bound anti-CD3 (1 μg/ml) and anti-CD28 (5 μg/ml) antibodies in R-10 media. On the following day, overnight-rested and activated CD8+ T cells were resuspended in DMEM-based XF media (Seahorse XF base media [Agilent Technologies Inc], 10 mM glucose, 1 mM sodium pyruvate, and 2 mM glutamine, pH 7.4) and plated by gentle centrifugation at 500,000 CD8+ T cells/well onto Cell-Tak (Cat. no. 354240; Corning) coated XFe96 plates as per Agilent protocols. Each biological sample had three to five technical repeats. Cells were then incubated at 37C for 30–60 min in a non-$CO_2$ XF Prep Station (Agilent) incubator and their metabolic profiles measured on a Seahorse XFe96 Analyzer initiated within 1 h of cell centrifugation. The optimum concentrations of mitochondrial inhibitors were determined experimentally to be 1 μM oligomycin A, 1.5 μM carbonyl cyanide 4-(trifluoromethoxy) phenylhydrazone (FCCP), 1 μM rotenone, and 1 μM antimycin A (#75351, #C2920, #R8875 and #A8674; Sigma-Aldrich).

### Flow cytometry CD98 and CD71 staining

Cryopreserved PBMCs were thawed, washed, and stained with Live/Dead Aqua Blue (Life Technologies), followed by surface staining with antihuman CD3 BV605, anti-human CD4 BV650, anti-human CD8 BV711, anti-human CD27 PerCP-eF710, CD45RO PE-CF594 (BD Biosciences), anti-human CD98 PE (BD Biosciences), anti-human CD71 APC, anti-human PD-1 PE-Cy7, and anti-human T-Bet BV421 (all from BioLegend unless otherwise noted). Anti-human CD14, anti-human CD16, and anti-human CD19 all tagged with V500 (BD Biosciences) were added in dump. For median fluorescence index determination, manual target channeling was performed with 1× beads (SpiroTech). Samples were run on modified 5 laser BD LSR Fortessa X-20. Data were analyzed on FlowJo (FlowJo).

### mTOR inhibition assay

PBMCs from freshly drawn blood were isolated over Ficoll–Hypaque gradient and stained with CFSE for 15 min at 37°C (Life Technologies). Half of the PBMCs, at 1.5 million/ml, were transiently exposed to rapamycin at 50 nM for an hour, washed and resuspended at 1.5 million/ml, and subsequently set up for 6-d proliferation assay with HIV gag, HIV nef, and CMV pp65 peptide pool along with SEB control at 100 ng/ml/peptide. The other half was set up for 6-d proliferation assay with HIV gag, HIV nef, and CMV pp65 peptide pool, along with SEB control at 100 ng/ml/peptide in the presence of 0, 12.5, 25, and

50 nM rapamycin. The 6-d proliferation assay was supplemented with 1 U/ml of rhIL-2 for cell viability. On the sixth day, the PBMCs were washed and set up for overnight cytokine secretion assay with the corresponding peptide pools along with SEB at 1 μg/ml/peptide in the presence of BFA at 1:1,000 dilution and co-stimulation antibodies anti-human CD28 and anti-human CD49d (BD Biosciences), both at 1:1,000 dilution. The following morning, PBMCs were washed, stained with Live/Dead Aqua Blue (Life Technologies), and surface-stained with anti-human CD3 BV605, anti-human CD8 BV711, anti-human CD4 APC, anti-human CD14 BV510, anti-human CD16 BV510, and anti-human CD19 BV510 (all from BioLegend). Afterwards, the cells were permeabilized with BD Cytofix/CytoPerm, washed, and stained with anti-human IFN-γ APC-Cy7 and anti-human TNF-α APC-Cy7 (all from BioLegend) to detect intracellular captured cytokines. Samples were run on modified 5 laser BD LSR Fortessa X-20. Data were analyzed on FlowJo (FlowJo).

### Killing assay

This method has been explained in detail elsewhere (Mujib et al, 2017). Briefly, negatively isolated $CD4^+$ T cells from cryopreserved PBMCs (according to the manufacturer's protocol, EasySep Human $CD4^+$ T-cell isolation kit; STEMCELL Technologies Canada Inc.) were activated on plate bound 1 μg/ml each of anti-human CD3 and anti-human CD28 (clones OKT3 and 28.2, respectively; BioLegend), and 50 IU/ml recombinant human IL-2 (rhIL-2; Roche Diagnostics) for 3 d before infecting with HIV-1 NL4-3 provirus. The killing assay was set up 3 d after infection with $CD8^+$ T cells and HIV-1–infected $CD4^+$ T cells at 1:1 ratio with 100,000 of each cell in the presence of 10 IU/ml rhIL-2 and the various inhibitors, if present. After overnight killing, the cells were stained with Live/Dead Aqua Blue (Life Technologies) and surface-stained with anti-human CD3 BV605, anti-human CD8 BV711, and anti-human CD4 APC (all from BioLegend). Afterwards, the cells were permeabilized and stained with anti–HIV-1 Gag p24 antibody Kc57-FITC (Beckman Coulter) to detect intracellular HIV-1 Gag protein. Samples were run on modified five laser BD LSR Fortessa X-20. Data were analyzed on FlowJo (FlowJo).

### Staining and detection of cGAPDH

Cryopreserved PBMCs were thawed and rested overnight in 37°C 5% $CO_2$ incubator. The following morning, $CD8^+$ T cells were negatively isolated (according to the manufacturer's protocol, EasySep Human $CD4^+$ T-cell isolation kit; STEMCELL Technologies Canada Inc.), stained with 75 nM MitoTracker Deep Red FM (Life Technologies) for 20 min at 37°C, and washed in 1× staining buffer (PBS with 1% FBS and 0.1% Sodium Azide). Afterwards, $CD8^+$ T cells were stained with CMV and HIV dextramers separately for 20 min in the dark and subsequently surface stained with anti-human PD-1 PE-Cy7 and anti-human CD8 AF594 (both from BioLegend) antibodies for an additional 20 min in the dark. $CD8^+$ T cells were washed in 1× staining buffer and fixed with 4% paraformaldehyde for 15 min at 37°C. $CD8^+$ T cells were washed again with 1× staining buffer and permeabilized on ice for 20 min in the dark with 300 μl of pre-chilled BD PhosFlow Perm Buffer III (BD Biosciences). After two extensive washes in 1× staining buffer, $CD8^+$ T cells were stained with 20 μg/ml Rabbit anti-GAPDH (Clone D16H11; Cell Signaling Technologies) in the dark for 15 min. $CD8^+$ T cells were washed again and then stained with 1.2 μg/ml of Goat anti-Rabbit antibody AF488 (SouthernBiotech) in the dark for 15 min. After washing again in 1× staining buffer, $CD8^+$ T cells were fixed in 1% PFA for 15 min at 37°C. After the final wash, stained $CD8^+$ T cells were resuspended in PBS with 1 μl/ml Hoeschst 33342 (BD Pharmingen) and data were collected on Amnis Image-Stream$^X$ Mk II Imaging Flow Cytomer. For each sample, at least 600,000 events were collected and analyzed on IDEAS software. All staining procedures were performed in 50 μl volume.

### Statistical analysis

Statistical analysis between the groups was assessed using GraphPad Prism. Specific tests for statistical significance are indicated in the figure legends. Differences were considered significant when *P*-values were below 0.05.

## Supplementary Information


## Acknowledgements

The authors wish to thank Leanne Wybenga-Groot of SPARC BioCentre (Molecular Analysis), The Hospital for Sick Children, Toronto, Canada, for assistance with Seahorse experiments. AKMN Rahman wants to thank Tasnuba Amin and members of Ostrowski lab for general support and discussion. This work was supported by Canadian Institutes of Health Research (CIHR) Fellowship to AKMN Rahman and funding from the Ontario HIV Treatment Network (OHTN) and CIHR.

### Author Contributions

AKMN Rahman: conceptualization, data curation, formal analysis, investigation, methodology, and writing—original draft, review, and editing.
J Liu: investigation and methodology.
S Mujib: data curation and investigation.
S Kidane: data curation and investigation.
A Arman: data curation and investigation.
S Szep: data curation and investigation.
C Han: data curation and investigation.
P Bonner: data curation and investigation.
M Parsons: data curation, formal analysis, investigation, and methodology.
E Benko: data curation.
CC Kovacs: data curation.
FY Yue: data curation.
M Ostrowski: conceptualization, formal analysis, supervision, funding acquisition, investigation, project administration, and writing—original draft, review, and editing.

### Conflict of Interest Statement

The authors declare that they have no conflict of interest.

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
