## [Reviewer comments · Life Science Alliance]

Life Science Alliance

Elevated Glycolysis Imparts Functional Ability to CD8+ T cells in HIV Infection

A.K. M. Nur-ur Rahman, Jun Liu, Shariq Mujib, Segen Kidane, Ali Arman, Steven Szep, Carrie Han, Phil Bonner, Michael Parsons, Erika Benko, Colin Kovacs, Feng Yun Yue, and Mario Ostrowski

DOI: <https://doi.org/10.26508/lsa.202101081>

Corresponding author(s): Mario Ostrowski, University of Toronto

Review Timeline:

Submission Date:	2021-03-30
Editorial Decision:	2021-06-14
Revision Received:	2021-08-25
Editorial Decision:	2021-08-27
Revision Received:	2021-09-03
Accepted:	2021-09-07

Transaction Report:

June 14, 2021

Re: Life Science Alliance manuscript #LSA-2021-01081-T

Prof. Mario Ostrowski
University of Toronto
1 King's College Circle, Rm 6271 Medical Sciences Building
Toronto, Ontario M5S 1A8
Canada

Dear Dr. Ostrowski,

Thank you for submitting your manuscript entitled "Elevated Glycolysis Imparts Functional Ability to CD8+ T cells in HIV Infection" to Life Science Alliance. The manuscript was assessed by expert reviewers, whose comments are appended to this letter. We invite you to submit a revised manuscript addressing the Reviewer comments.

Thank you for this interesting contribution to Life Science Alliance. We are looking forward to receiving your revised manuscript.

Sincerely,

- A letter addressing the reviewers' comments point by point.
- An editable version of the final text (.DOC or .DOCX) is needed for copyediting (no PDFs).
- High-resolution figure, supplementary figure and video files uploaded as individual files: See our detailed guidelines for preparing your production-ready images, <https://www.life-science-alliance.org/authors>
- Summary blurb (enter in submission system): A short text summarizing in a single sentence the study (max. 200 characters including spaces). This text is used in conjunction with the titles of papers, hence should be informative and complementary to the title and running title. It should describe the context and significance of the findings for a general readership; it should be written in the present tense and refer to the work in the third person. Author names should not be mentioned.

B. MANUSCRIPT ORGANIZATION AND FORMATTING:

Reviewer #1 (Comments to the Authors (Required)):

Elevated Glycolysis Imparts Functional Ability to CD8+ T cells in HIV Infection

A K M Nur-ur Rahman

Neuro-immunometabolism in the new frontier in research in cancer and chronic viral infections where both Warburg effect and OxPHOS are at play during intense demand for immune fight. Increasing evidence indicates that immune cells rewire their metabolism in the presence of persisting antigen. The high intracellular levels of lactate and reactive oxygen species (ROS) generated during enhanced aerobic glycolysis and mitochondrial oxidative phosphorylation respectively led to oxidative stress. The detoxification of these accumulating metabolites and the equilibrium between reduced and oxidized nicotinic adenine dinucleotide (NADH and NAD+) are two prominent mechanisms regulating redox status and hence energy homeostasis in cancer, COVID and HIV infections. Schurich et al 2016.

Bergamaschi L, Longitudinal analysis reveals that delayed bystander CD8+ T cell activation and

early immune pathology distinguish severe COVID-19 from mild disease. *Immunity*. 2021 May 16;S1074-7613(21)00216-8.

Jung JG, Le A. Metabolism of Immune Cells in the Tumor Microenvironment. *Adv Exp Med Biol*. 2021;1311:173-185. à

Investigators assessed functional metabolic in CD8⁺ T cells during acute, chronic HIV infection in elite controllers. They found that ex vivo CD8⁺ T cells at maximal OXPHOS capacity even at rest while mitochondrial energy demands are high during early untreated and chronic infection, as well as in states of excellent viral control.

CD8⁺T cell cytoplasmic GAPDH content was reduced in HIV, but less in early infection and viral controllers. Thus, CD8⁺T cell exhaustion in HIV is characterized by reduced glycolytic activity, enhanced OXPHOS demands, dysregulated mTOR, and reduced cytoplasmic GAPDH .

However important issues have to be addressed

Introduction and discussion are not enough focus on update of immunometabolism in HIV and should directly present objective to address new question following accepted evidence from literature.

Passaes C., Optimal Maturation of the HIV-Specific CD8 T Cell Response after Primary Infection Is Associated with Natural Control of HIV: ANRS SIC Study. *Cell Rep*. 2020 Sep 22;32(12):108174..

Sáez-Cirión A, Sereti I. Immunometabolism and HIV-1 pathogenesis: food for thought. *Nat Rev Immunol*. 2021 Jan;21(1):5-19.

Angin M, Metabolic plasticity of HIV-specific CD8⁺ T cells is associated with enhanced antiviral potential and natural control of HIV-1 infection. *Nat Metab*. 2019 Jul;1(7):704-716.

Data generated are difficult to interpret for the reduced glycolytic activity as elevated SUV glucose metabolism in HIV patients is elevated by PET scan and normalized on ART after at least 6 months of therapy.

Brust D., Fluorodeoxyglucose imaging in healthy subjects with HIV infection: impact of disease stage and therapy on pattern of nodal activation. *AIDS*. 2006 Apr 24;20(7):985-93 and 2006 Feb 28;20(4):495-503.

Immunometabolism data combined with exhaustion markers in different groups acute chronic viral control and in different cell subsets making study objectives confused and hard to follow. The direct comparison between HIV specific and CMV specific immunometabolism changes will be of great interest as total CD8 may represent bystander activated cells and CD8 may represent a very heterogeneous groups (bystander vs. specific for HIV and CMV. Access to patients collected by leukapheresis who large number of PBMCs may help to dig in this key objective.

Rapamycin and metformin functional assays will be of interest.

Discussion investigators should discuss on potential specific intervention based on their immunometabolism discovery.

Reviewer #2 (Comments to the Authors (Required)):

This manuscript describes evaluations of metabolic activities in CD8 T cell from HIV-infected patients, relative to healthy control persons. A large number of experimental findings are described, including assessments of metabolic activities ex vivo and after in vitro culture, functional killing assays, and metabolic profiles after experimental manipulation of CD8 T cell manipulation with specific chemicals or pharmacological agents. Exploring the metabolism of CD8 T cells in HIV infection is not entirely novel, and the findings presented here are frequently not particularly strong. In addition, this manuscript could easily be shortened by focusing on a smaller number of key

findings, while de-prioritizing the non-significant findings (e. g. Figure 6, Figure 8, Figure 9). Specific comments:

1. Differences in Figure 1B-H, although significant in some cases, are frequently minor and their physiological significance is uncertain.
2. Figure 2I-P could be provided in a supplement figure.
3. Figure 5B-G describes phenotypic changes of CD8 T cells, but the functional connection between these phenotypic markers and metabolic activities is not experimentally addressed. In the absence of such functional evaluations, the significance of the described phenotypic differences remain uncertain.
4. Figure 7: I recommend to move all non-significant data to supplements. Effects of DCA on killing activities of CD8 T cells could be explored in more detail.
5. Both the results section and the discussion are very long. Overall, I believe the key findings of this experimental work can be summarized from a more high-level perspective in 3-4 pages.
6. Generally, I believe the findings reported here are of moderate interest and significance.

The Editor
Life Science Alliance

Dear Editor,

I am enclosing a copy of a first revised versions of the manuscript entitled, “Elevated Glycolysis Imparts Functional Ability to CD8+ T cells in HIV Infection”, by Rahman et. al, for consideration of publication as a research article in Life Science Alliance.

We thank the Reviewers for their helpful comments. The following are responses to the Reviewers, with changes/revisions highlighted yellow in the Revised version of the manuscript.

Reviewer #1:

“Neuro-immunometabolism in the new frontier in research in cancer and chronic viral infections where both Warburg effect and OxPHOS are at play during intense demand for immune fight. Increasing evidence indicates that immune cells rewire their metabolism in the presence of persisting antigen. The high intracellular levels of lactate and reactive oxygen species (ROS) generated during enhanced aerobic glycolysis and mitochondrial oxidative phosphorylation respectively led to oxidative stress. The detoxification of these accumulating metabolites and the equilibrium between reduced and oxidized nicotinic adenine dinucleotide (NADH and NAD⁺) are two prominent mechanisms regulating redox status and hence energy homeostasis in cancer, COVID and HIV infections. Schurich et al 2016.

Bergamaschi L, Longitudinal analysis reveals that delayed bystander CD8+ T cell activation and early immune pathology distinguish severe COVID-19 from mild disease. Immunity. 2021 May 16;S1074-7613(21)00216-8.

Jung JG, Le A. Metabolism of Immune Cells in the Tumor Microenvironment. Adv Exp Med Biol. 2021;1311:173-185. À”

We recognize that redox status is another aspect of T cell metabolism for which we did not address in this manuscript, and thus is beyond the scope of the current manuscript but

should be addressed in future studies. Our main focus for the current work was to systematically examine human HIV specific T cell metabolism using Seahorse respiration technology at all clinical states of this infection.

“Introduction and discussion are not enough focus on update of immunometabolism in HIV and should directly present objective to address new question following accepted evidence from literature.

Passaes C., Optimal Maturation of the

SIV-Specific CD8 T Cell Response after Primary Infection Is Associated with Natural Control of SIV: ANRS SIC Study. Cell Rep. 2020 Sep 22;32(12):108174..

Sáez-Ciri3n A, Sereti I. Immunometabolism and HIV-1 pathogenesis: food for thought. Nat Rev Immunol. 2021 Jan;21(1):5-19.

Angin M, Metabolic plasticity of HIV-specific CD8⁺ T cells is associated with enhanced antiviral potential and natural control of HIV-1 infection. Nat Metab. 2019 Jul;1(7):704-716.”

We have now added the Passaes and Angin paper references in the Introduction (lines 79 and 88) and also discuss the Angin findings in the Discussion section (line 528). The Angin paper primarily examined transcriptomics (ie RNA levels) whereas we focused on functional readouts of respiration (seahorse data). In the Angin et al paper, the authors compared untreated viral controllers with cART treated individuals. HIV specific CD8⁺ T cells decay while on cART (PMID: 9516110 Ogg et al Science 1989) and thus PBMC from these individuals may have dramatically lower frequencies of HIV specific cells than untreated individuals, thus comparing suppressive activity is confounded by differences in HIV specific CD8s in the two groups found in PBMC samples. A more relevant comparison would have been comparing viral controllers with untreated viremic progressors, which we do in our study. We could not see enhanced mTOR activity ex vivo in acute and chronic HIV but normal levels of mTOR activity after activation through the TCR by CD3. The Angin paper looked at mTOR activation after co-culture with HIV peptides, and in that experimental condition, mTOR was activated. These findings are consistent with our data with CD3 activation in which there was a ‘tendency’ to enhanced mTOR after activation, thus our findings are consistent with the Angin et al study.

Immunometabolism is a complex topic in which the general readership is not very familiar with, and thus we tried to articulate the assays and their interpretations with lengthy and hopefully clear explanations. We intended to limit Discussion mainly relevant to the studies that we performed.

“Data generated are difficult to interpret for the reduced glycolytic activity as elevated SUV glucose metabolism in HIV patients is elevated by PET scan and normalized on ART after at least 6 months of therapy. Brust D., Fluorodeoxyglucose imaging in healthy

subjects with HIV infection: impact of disease stage and therapy on pattern of nodal activation. AIDS. 2006 Apr 24;20(7):985-93 and 2006 Feb 28;20(4):495-503.”

PET scanning directly looks at high metabolic activity, such as glucose uptake rather than glycolysis, per se, and the cell populations involved in PET scanning will also include B cells, CD4 cells, and macrophages, that is, any cell upregulating Glut1. We did see elevated glycolysis in untreated acutes and viral controllers but only slight increases in glycolysis of ex vivo CD8 T cells in untreated chronics (see Fig 2G). The CD8s in untreated chronics are clearly activated (elevated CD71 in Fig 5E and PD1 in Fig 9). We postulate that increased numbers of immune cells, with mild but probably insufficient increases in glycolysis as seen in chronic infection would result in measurable FDG uptake due to high cell numbers in lymph nodes that have active HIV replication. Also, the Brust et al study noted by Reviewer #1 above is difficult to interpret; for eg. In their Table 2, 3 out of 6 control participants and 2/3 LTNP had increased F-deoxyglucose uptake in lymph nodes which does not support the main hypothesis of that paper.

“Immunometabolism data combined with exhaustion markers in different groups acute chronic viral control and in different cell subsets making study objectives confused and hard to follow.”

In Fig 9 we showed exhaustion marker data to indicate the degree of PD1 expression on CD8s in our different subject groups. The levels of exhaustion are what have previously been reported by others.

“The direct comparison between HIV specific and CMV specific immunometabolism changes will be of great interest as total CD8 may represent bystander activated cells and CD8 may represent a very heterogenous groups (bystander vs. specific for HIV and CMV. Access to patients collected by leukapheresis who large number of PBMCs may help to dig in this key objective. Rapamycin and metformin functional assays will be of interest.”

Our data was based on obtaining leukapheresis samples. This is because Seahorse technology requires large numbers of cells, and with different conditions, one can only do these experiments using leukapheresis sampling. It would be very difficult to do our seahorse assays by sorting out CMV and HIV specific cells by tetramer/dextramer technology due to the high cell numbers that would be required; which may be beyond what can be obtained from leukapheresis samples, and also be prohibitively expensive to perform and beyond our research budget. We did attempt to look at virus specific cells with our GAPDH studies and we find an overall bystander effect to our surprise. We performed extensive rapamycin experiments depicted in Figure 6. We did not use metformin in our assays as metformin effects appear to be pleiotropic being associated with AMPK dependent and AMPK independent effects (see PMID: 25456737), thus we felt interpretation would be problematic in this regard.

“Discussion investigators should discuss on potential specific intervention based on their immunometabolism discovery.”

Again, we did not want to have a lengthy Discussion, especially with concerns from Reviewer #2 below, who believes our Discussion is too long. We do discuss potential interventions that should be tested in the future in the final paragraph “In future studies, metabolic therapeutics that can enhance glycolytic capacity, and mTOR1 activation should be considered along with effective cART to recuperate metabolic functional profiles of CD8+ T cells to maximize their functional control of HIV.”

Reviewer #2:

“Exploring the metabolism of CD8 T cells in HIV infection is not entirely novel, and the findings presented here are frequently not particularly strong. In addition, this manuscript could easily be shortened by focusing on a smaller number”

Again, this is the first study to our knowledge to systematically examine CD8 T cell metabolism for all HIV clinical states using seahorse technology; in addition our GAPDH studies have not been done before in HIV infection; our rapamycin experiments further define the role of mTOR in HIV Infection, and to our knowledge this is one of the few studies looking at mechanisms of CD8 killing in human samples attempting to tease out contributions of glycolysis versus OXPHOS. We have considerably shortened the Results and Discussion section as requested below but have kept the explanations of the assays in the text, as metabolic studies are not often understood well by the general readership. We have now moved the Table 1 as a Supplemental Table 1.

Specific comments:

1. Differences in Figure 1B-H, although significant in some cases, are frequently minor and their physiological significance is uncertain.

Seahorse technology as depicted in Figures 1-4 generally has a narrow dynamic range particularly when comparing respiration and glycolysis on direct ex vivo human samples or after one round of T cell activation. Minor significant changes may translate to major differences in a chronic infection where there are multiple rounds of continuous activation in response to antigen. One observes similar differences when examining the literature (for eg, see Angin et al Nat Metab. 2019 Jul;1(7):704-716 where T cells are stimulated with IL15 (fig 8 b) one sees modest differences compared to unstimulated and yet this would presumably be the maximal response.)

“Figure 2I-P could be provided in a supplement figure”

We have now modified Figs 1-3 to make them more easily readable and inserted most of the overnight activated data into supplementals.

“Figure 5B-G describes phenotypic changes of CD8 T cells, but the functional connection between these phenotypic markers and metabolic activities is not experimentally addressed. In the absence of such functional evaluations, the significance of the described phenotypic differences remain uncertain.”

The main purpose of the Fig 5 data is to determine whether mTOR activation is associated with enhanced glycolytic activity, which was not the case. We have now modified Fig 5 to only show relevant information. That is, CD8s are activated during chronic infection, however, this is not associated with mTOR activation. We have now shortened this section and removed Fig 5D and 5G.

“Figure 7: I recommend to move all non-significant data to supplements. Effects of DCA on killing activities of CD8 T cells could be explored in more detail.”

We feel it is important to demonstrate crucial non-significant findings when blocking certain pathways which would demonstrate that blocking that pathway has no role in killing effects. The data show that killing is mainly mediated by glycolysis (Fig 7D and G) for which we find the effect mainly with DCA exposure during the killing assay, but also a trend to an effect even with pre-treatment with DCA. We are uncertain of what further work should be explored regarding DCA, since DCA is blocking glycolysis and enhancing OXPHOS. We cannot completely eliminate a role for OXPHOS in killing activity since we see reduction of killing in presence of oligomycin 7/10 individuals (7E).

“Both the results section and the discussion are very long. Overall, I believe the key findings of this experimental work can be summarized from a more high-level perspective in 3-4 pages”

We agree the paper is lengthy due to the number of assays performed and on samples from all HIV clinical states. We have now reduced the Results and Discussion sections to be more concise. Table 1 and many of the original figures are now in supplemental.

“Generally, I believe the findings reported here are of moderate interest and significance.”

We thank the Reviewer for this encouraging comment!

Sincerely yours,

Mario Ostrowski, M.D.,

August 27, 2021

RE: Life Science Alliance Manuscript #LSA-2021-01081-TR

Prof. Mario Ostrowski
University of Toronto
1 King's College Circle, Rm 6271 Medical Sciences Building
Toronto, Ontario M5S 1A8
Canada

Dear Dr. Ostrowski,

Thank you for submitting your revised manuscript entitled "Elevated Glycolysis Imparts Functional Ability to CD8+ T cells in HIV Infection". We would be happy to publish your paper in Life Science Alliance pending final revisions necessary to meet our formatting guidelines.

- please upload your main manuscript text as an editable doc file
- please upload your main and supplementary figures as single files
- please upload your Tables in editable .doc or excel format
- please add the Twitter handle of your host institute/organization as well as your own or/and one of the authors in our system
- please be sure that all authors are inserted in the Author Contribution section in the manuscript text
- please add a conflict of interest statement to your main manuscript text
- please use the [10 author names, et al.] format in your references (i.e. limit the author names to the first 10)
- please add your main, supplementary figure, and table legends to the main manuscript text after the references section
- please add call-outs for Figures 4B, S1A-B, S2A-B, S3A-B, S4A-F to your main manuscript text

LSA now encourages authors to provide a 30-60 second video where the study is briefly explained. We will use these videos on social media to promote the published paper and the presenting author. Corresponding or first-authors are welcome to submit the video. Please submit only one video per manuscript. The video can be emailed to contact@life-science-alliance.org

A. FINAL FILES:

B. MANUSCRIPT ORGANIZATION AND FORMATTING:

Sincerely,

September 7, 2021

RE: Life Science Alliance Manuscript #LSA-2021-01081-TRR

Prof. Mario Ostrowski
University of Toronto
Medicine
1 King's College Circle, Rm 6271 Medical Sciences Building
Toronto, Ontario M5S 1A8
Canada

Dear Dr. Ostrowski,

Thank you for submitting your Research Article entitled "Elevated Glycolysis Imparts Functional Ability to CD8+ T cells in HIV Infection". It is a pleasure to let you know that your manuscript is now accepted for publication in Life Science Alliance. Congratulations on this interesting work.

DISTRIBUTION OF MATERIALS:

Again, congratulations on a very nice paper. I hope you found the review process to be constructive and are pleased with how the manuscript was handled editorially. We look forward to future exciting submissions from your lab.

Sincerely,
